# Flow-Type Landslides Analysis in Arid Zones: Application in La Chimba Basin in Antofagasta, Atacama Desert (Chile)

**Francisca Roldán** [1,2,*], **Iván Salazar** [3], **Gabriel González** [1,2], **Walter Roldán** [3] and **Norman Toro** [4]

1   Departamento de Ciencias Geológicas, Universidad Católica del Norte, Antofagasta 1270709, Chile; ggonzale@ucn.cl
2   Research Center for Integrated Disaster Risk Management (CIGIDEN), Santiago 7820436, Chile
3   Departamento de Ingeniería Civil, Universidad Católica del Norte, Antofagasta 1270709, Chile; isalazar@ucn.cl (I.S.); wroldan@ucn.cl (W.R.)
4   Faculty of Engineering and Architecture, Universidad Arturo Prat, Iquique 1100000, Chile; notoro@unap.cl
*   Correspondence: francisca.roldan@cigiden.cl; Tel.: +56-956479506

**Abstract:** In the world, the hazards of intense rainfall are recurrent and increasing. In addition, they are one of the natural hazards that cause the most severe damage to infrastructure and even cause deaths every year. Flow-type landslides are capable of develop in areas with different geomorphological, geological and climatic characteristics. In hyper-arid zones such as the Atacama Desert, these hazards are capable of develop in a timely manner, causing catastrophes. This study analyzes the flow-type landslide in a hyper-arid mountainous area in La Chimba basin of Antofagasta city (Chile). For this, a hydrometeorological analysis is carried out through a pluviometric analysis, statistical analysis of frequencies through the Gumbel probabilistic method of extreme values and determination of maximum flows by obtaining IDF (intensity-duration-frequency) curves and design rainfall intensity as a function of concentration time. To obtain the maximum flows of liquid runoff and debris, for different return periods, the rational method was used with the method proposed by O'Brien. For the determination in the impact zone, the modeling software HEC-RAS (Hydrologic Engineering Center's River Analysis System) and RAMMS (Rapid Mass Movements). Hydrographs are used for a return period of about 200 years, considered the most unfavorable scenario with the Voellmy–Salm model. To validate the modeling, a morphometric, sedimentological and comparative analysis is carried out between real impact zones of 1991 event and those generated in this study. It is concluded that the sedimentological and morphometric characteristics indicate that the type of flow that it can originate would have a rapid response to rainfall events of great intensity or duration. The modeling provided by HEC-RAS represents a fluvial-type flow, while the RAMMS modeling is closer to the consistency of a flow-type landslide, which is estimated to be closer to reality. The results show that despite being in a hyper-arid zone, the rainfall factor is capable of landslides triggering in mountainous areas.

**Keywords:** landslides; debris flow; hydrometeorology; alluvial; hyper-arid; hazard

## 1. Introduction

On a global scale, climate change has caused enormous damage, in particular, flow-type landslides. Only from the year 2000 to 2019, 7348 of them have been registered, increasing by 174% compared to the 20 previous years (1980–1999). This caused 1.23 million deaths and USD 2.97 trillion in economic losses and affected over 4 billion people, with a surge in the number of climate-related disasters, of which floods were the most frequent, accounting for 44 % of all disasters. In fact, of various hazard generated by natural phenomena, hazards from intense rainfall are recurrent and on the rise [1,2]. Specifically the landslides are one of the most frequent geological hazards in mountainous regions [3]. They are one of the natural hazards that cause the most severe damage to infrastructure

and even cause deaths every year. They also represent the most relevant geomorphological processes in the construction and modification of the relief [4–7].

Varnes [8] defined landslides as a downward and outward movement of materials that make up the slopes composed of natural rock, soils, artificial fills or combinations of these materials. On the other hand, Hungr et al. [9] gives an updated definition of Varnes [8] defining flow-type landslides as flows of saturated debris that arise from very fast to extremely fast and that occur in steep channels with a significant drag of material and water. The difficulty of its study lies in the fact that its origin is due to a multiplicity of climatic and non-climatic factors that act directly and indirectly in a given space and time [10], such as the topographic, geological, hydrographic, geotechnical, vegetation and land use factor (e.g., [8,11–16]). However, according to Pánek [17], climatic factors represent some of the most important causes and triggers of landslides. In addition, due to recent extreme rain events around the world, damage caused by flow-type landslides frequently occurs due to long-duration and/or heavy rainfall, resulting in significant human and material losses [18].

Specifically in Chile, landslides generated by hydrometeorological processes are one of the most frequent hazards [1,19–22]. In addition, Chile geography is favorable for its development [19,21] due to the extensive hydrographic basins developed mainly in the Coastal Range and Andes Range. Moreover, climate change records indicate that the frequency of rainfall has decreased and will decrease in Chile, but will increase in intensity [23]. As a consequence, torrential events are likely to occur that will generate abundant hazards and extensive landslides impact areas. This is important, considering that rainfall is the main "trigger" for landslides [24]. To reduce the impact of flow-type landslide events, impact areas must be known, especially in areas where they are highly urbanized. This is fundamental, considering that at the country situation; Chile does not have a monitoring and early warning system, counting only on rain alerts in a territory where urban expansion is not controlled. This include countless informal homes (slums), built with little resistant materials and even waste that are positioned in impact landslides areas such as the Coastal Range foothills. In addition, a considerable number of landslide events have developed in the territory, such as debris flows with catastrophic consequences, thus demonstrating the vulnerability of this territory to this type of natural hazard.

The studies developed, in the world, most analyzing of rain-induced landslides have been focused on humid regions [25]. However, little is known about landslide occurrence in arid and semi-arid environments, and we do not have much information about their downstream impacts and their importance within the global sediment balance [26]. Furthermore, according to Moreiras and Sepúlveda [27], the exclusive sceneries of arid landscapes force a complex environmental mechanism, that lead to a series of particular landforms and landslides processes that may vary from place to place.

Antofagasta city (Chile) is a case of this. It is in the Atacama Desert, and despite being in a hyper-arid climate has presented torrential rainfall events that have triggered the activation of the basins located in the Coastal Range, causing the death of civilians and considerable economic damage. It currently has a lack of control in the settlement of houses in the foothills without counting on an early warning system and with precarious evacuation strategies. In addition to this, in this locality there are no studies that determine the impact zones and neither determine the hydrological response of the basin in this hyper-arid zone. The existing studies are technical reports and analysis of hydrometeorological events of the "Servicio Nacional de Geología y Minería" (e.g., [23,28]) in addition to some publications focused on the meteorological analysis of the phenomena (e.g., [29]) and the description of some flow-type landslide events (e.g., [29,30]). Although it is information that contributes to the knowledge of the characteristics of these landslide phenomena, it is still uncertain whether the existing methodologies for the determination of impact zones and hydrological response of the basins are valid for this hyper-arid location of Chile.

Considering the above and that the most important active processes in arid environments are related to stream run off triggered by extreme rainfall events. Even low-intensity

rainfall may increase ephemerally surface runoff with an extraordinary erosive power [27]. It is essential to predict the flow-type landslides impact areas to establish adequate territorial planning and to prevent socio-natural disasters from developing again. This prediction is often not possible due to limited geological information or details about triggering mechanisms, such as an extreme rainfall event [31]. Therefore, the so called retrospective analysis of past debris flow events or historical rainfall for the calibration of digital models can be used to design risk reduction measures [32] and for the generation of impact maps. In this study, mathematical simulation models of the debris flow displacement are carried out through the HEC-RAS 6.1 (U.S. Army Corps of Engineers (USACE), Washington, DC, USA) and RAMMS v1.7 (WSL Institute for Snow and Avalanche Research SLF, Davos, Switzerland) software. For its calibration, a return period of 200 years considered the worst scenario is completed by a record of historical and recent rains for the La Chimba basin of Antofagasta city (Chile), in addition to morphometric, soil analysis and retrospective analysis for validation result.

Considering the uncertainty of climate change with respect to future rainfall, in addition to the inability to analyze the deposits of alluvial fan in urban areas, it is that the modeling of flow-type landslides to determine the areas of impact are fundamental for disaster risk mitigation. Especially considering that in arid and semi-arid complex environments display particular characteristics with an ensemble of different factors that can facilitate slope failures and mass movement. Poor soil development, intense physical weathering, intense storms, droughts and wind particles deposition in these climatic environments provide the stage on which the landslides are either the end results thereof or active participants of land denudation [27].

*Study Area*

Antofagasta city is in the Atacama Desert of northern Chile. Specifically in a narrow strip located between the Coastal Escarpment, with more than 50 hydrographic basins of various surfaces located in the Coastal Range that flow into urban areas (Figure 1). It is characterized by the development of abundant low cloudiness ("camanchaca"), an average daily temperatures between 13 °C (winter) and 20 °C (summer) and an average annual rain of 3 to 4 mm. Despite being in an area of hyper-arid climate, it is sporadically affected by convective storms that can generate alluvial runoff of great magnitude [33] (Figure 2) that have caused debris and/or mud flows, causing loss of human lives, extensive damage to homes and urban infrastructure [33,34]. The development of flow-type landslides responds to specific events that develop in case of climatic anomalies such as the El Niño phenomenon, or others [21,28,33] (Figure 2). This phenomenon corresponds to the warm phase of the Southern Oscillation (ENSO), constitutes an extreme state in the ocean atmospheric conditions of the Pacific basin [33].

Antofagasta records the largest flow-type landslide disaster in Chile, which occurred on 18 June 1991 (Figure 2). This event caused 91 deaths, 19 missing bodies, 70,000 affected and 6000 homes destroyed [34] with a total variable rainfall record, between 14 and 42 mm for a duration close to 3 h, being higher than the annual historical average recorded from 1919 to date (4.4 mm) (DMC). Because of this catastrophic event, public policies for the construction of alluvial mitigation were established. Despite this, the absence of urbanization at that time left several basins of the northern sector of the city, a sector in full urban development, which is currently urbanized and without alluvial mitigation works.

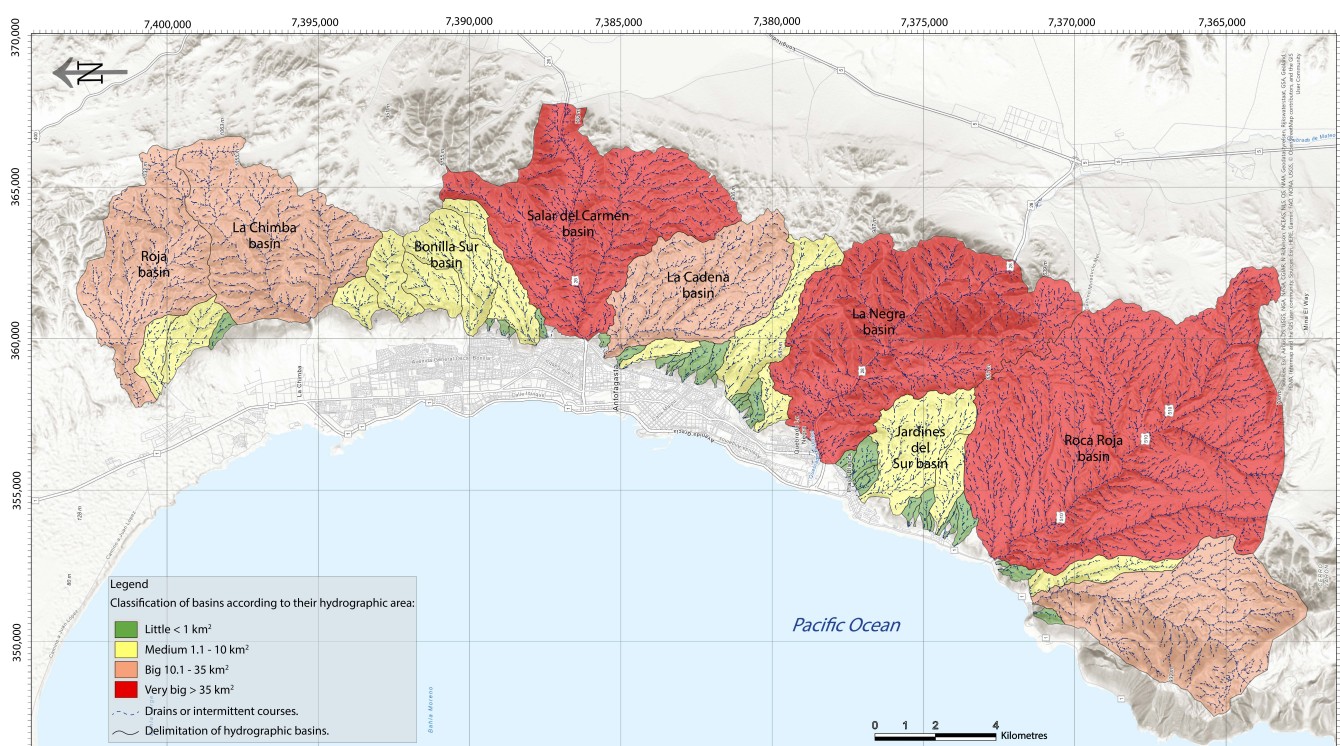

**Figure 1.** Basins of the city of Antofagasta. The colors indicate a classification according to its hydrographic area.

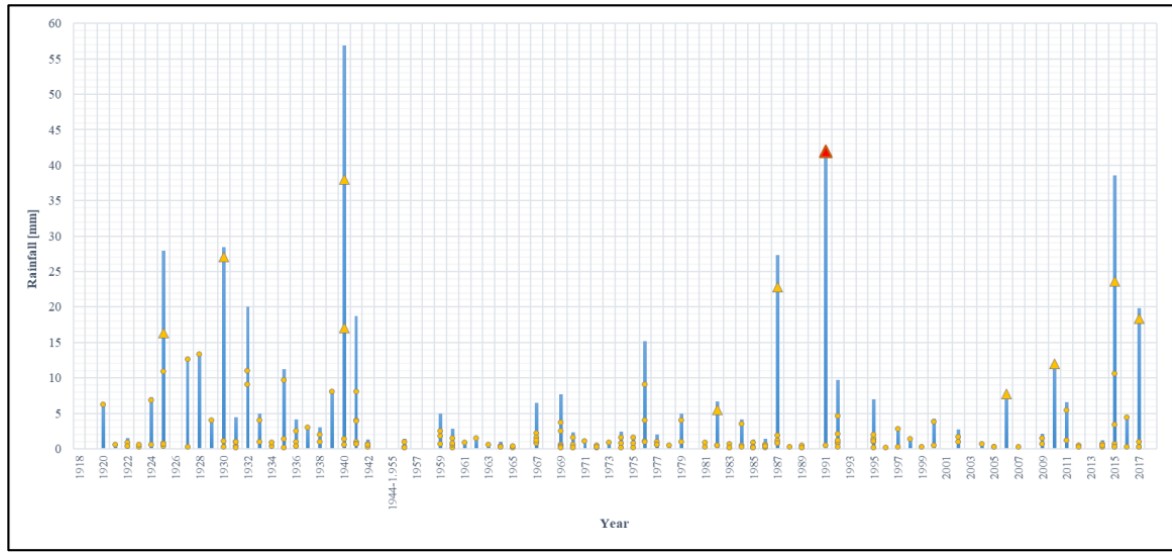

**Figure 2.** Record of annual (blue bar) and daily (yellow circles) rainfall between the years 1918–2018 of the city of Antofagasta. The yellow triangles indicate rainfall with flooding and/or development of flow-type landslides. The red triangle indicates the hydrometeorological of the flow-type landslide event which occurred in Antofagasta city on 18 June 1991. This event is considered the largest landslide disaster registered in Chile. Information source: [33,34]. Rainfall records from the Dirección Meteorológica de Chile (DMC).

Specifically, the study area is in the northern sector of the city of Antofagasta, between the coordinates 23°30′ and 23°34′ South latitude and 70°17′ and 70°24′ West longitude; bounded to W by the coastal edge and to E by the trace of the Mititus fault, corresponding to the Atacama Fault System (Figure 3A,B). This area is made up of the La Chimba basin

with an area of 24.7 km² (Figure 3C), being considered one of the largest basins in the city of Antofagasta (Figure 1); in addition, according to Araya [35] and Chong et al. [36] there is previous evidence of activations, as in the hydrometeorological event of 1991. Despite the background presented, currently, the alluvial deposits of the mouth of the La Chimba basin have a significant number of urbanizations, located adjacent to the La Chimba dump and an aggregate extraction site. Likewise, real estate projects will be developed, projecting a significant population increase. Considering that this basin is gaining importance, there are no studies that estimate the type of hydrological response to possible rainfall events, nor an estimate of the possible areas of impact for urban areas and their future growth, nor the consideration of how flow-type landslides mitigation works.

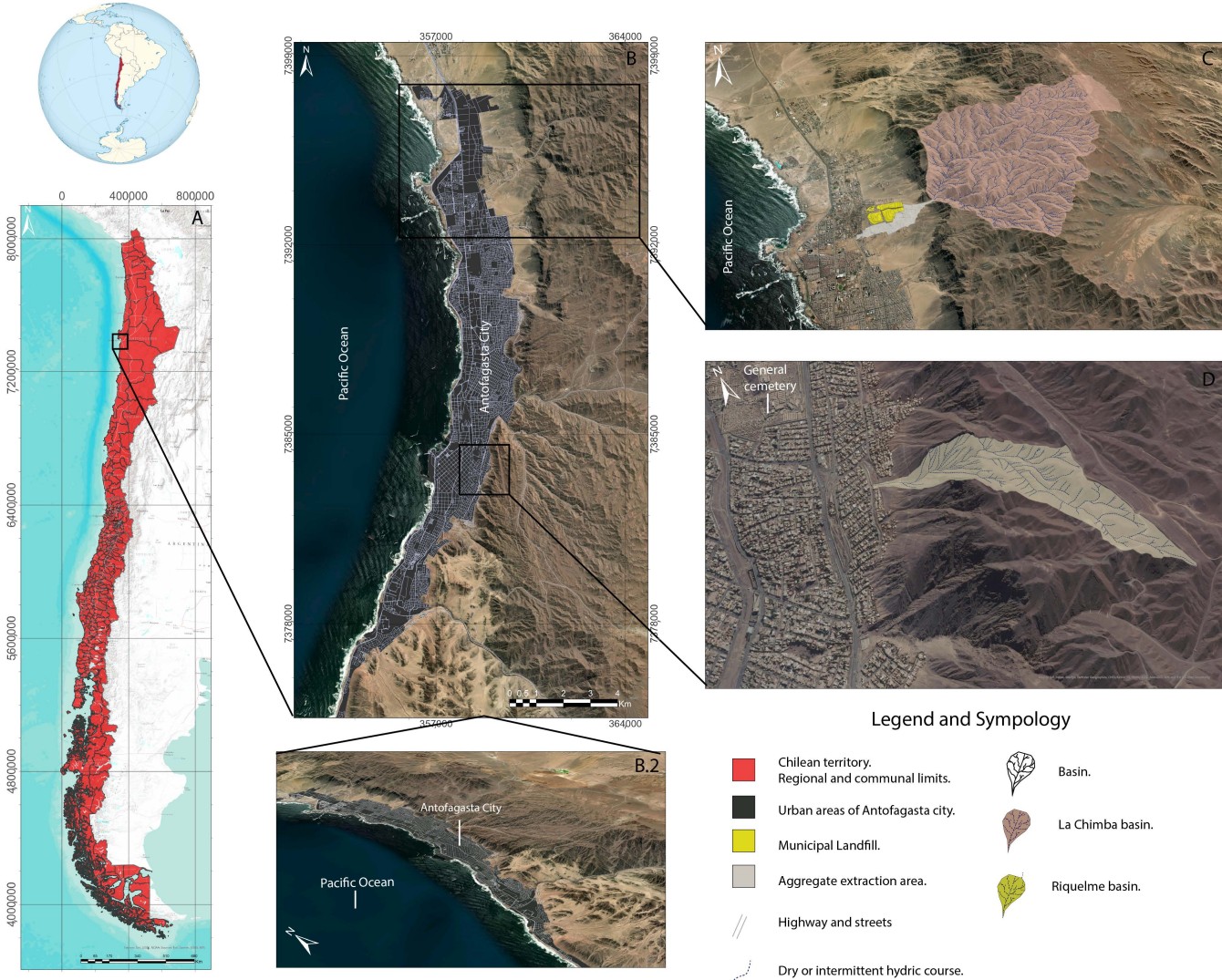

**Figure 3.** Location map. (**A**) Indicates the location of Chile on a country scale. (**B**) Indicates the location of the city of Antofagasta, Antofagasta Region. (**B.2**) Indicates the location of the city of Antofagasta with a 3D view. (**C**) It is the location of the La Chimba basin with a 3D view. (**D**) It is the location of the Riquelme basin with a 3D view (validation method).

## 2. Methodology

### 2.1. Hydrometereological Characterization

The objective is to determine the flow-type landslide impact areas with an application in La Chimba basin. A mathematical modeling of the progress of this hazard was carried out with the use of HEC-RAS v6.1 (U.S. Army Corps of Engineers (USACE), Washington, DC, USA) and RAMMS v1.7.20 software ((WSL Institute for Snow and Avalanche Research SLF, Davos, Switzerland).

For this, 4 previous procedures were developed: pluviometric analysis, frequency statistical analysis, IDF curves calculation (intensity-duration-frequency) and design rainfall intensity as a function of concentration time. Finally, the maximum liquid ($Q_L$) and detrital ($Q_D$) calculation runoff flows were carried out for the different return periods.

#### 2.1.1. Pluviometric Analysis

There are defined the meteorological stations that represent the best way the rainfall events that have occurred in the study areas through the historical recompilation from data. This allows an estimation of design rainfall closer to reality and, therefore, the debris flows ($Q_D$) to be considered.

Included were: institution in charge of the measurements, name of the station, coordinates (location), height of measurement and range of years with data availability. After this, the meteorological station to be used was selected considering orographic effects in the registers, wind effects, problems due to exposure, distance of the meteorological stations in relation to the study area, years with/without pluviometric record and differences in rainfall records between stations.

#### 2.1.2. Frequency Statistical Analysis and Maximum Flow Rates Calculation

The maximum flow-type landslide for a specific return period ($Tr$) and time of concentration ($T_c$) were obtained through a frequency statistical analysis and the calculation of exceedance and non-exceedance probabilities, in addition to rainfall for a specific return period ($Tr$). Next, the maximum liquid of flow-type landslide, through the IDF curves, frequency coefficient for a return period, design rainfall intensity for a return period ($Tr$—years), flow maximum liquid $Q_L$ (m$^3$/s) using the rational method and the maximum debris flow ($Q_D$) were calculated using the O'Brien and Julien (1997) method based on the concentration time calculated [37–42] (Appendix A).

#### 2.1.3. Flow-Type Landslide Mathematical Modeling

There are several mathematical equations that control the modeling of the HEC-RAS and RAMMS software, in addition to their corresponding parameters to insert, highlighting some of them such as: flow height in a non-permanent flow regime ($H$), roughness coefficient of Manning ($n$), total friction ($S_f$), volume of solids in the flow ($V_d$) and average debris flow velocity ($V_m$), among others (Appendix B).

Moreover, in RAMMS, the total basal friction in the Voellmy–Salm model was divided into a dry Coulomb friction coefficient $\mu$, and a turbulent friction coefficient $\xi$ (m/s$^2$). From these values, different scenarios were established for mathematical modeling, with the aim of representing different scenarios to obtain a flow of maximum, average and minimum fluidity (Table 1).

**Table 1.** Values considered for flow-type landslide mathematical models with variation in the state of fluidity. Source: values from [43].

| Fluidity | Parameter | Value |
|---|---|---|
| Maximum fluidity | $\xi$ (m/s$^2$) | 500 |
| | $\mu$ | 0.05 |
| Medium fluidity | $\xi$ (m/s$^2$) | 350 |
| | $\mu$ | 0.125 |
| Minimum fluidity | $\xi$ (m/s$^2$) | 200 |
| | $\mu$ | 0.2 |

In addition, the flow density values were considered based on the study by Costa [44] in Ayala et al. [42], where it was estimated from the rheological-hydrodynamic point of view that flow-type landslides, considered non-Newtonian Viscoplastics present densities between 1.8 and 2.3 gr/cm$^3$, where in this particular study a density of 2.0 gr/cm$^3$ is considered as a standard measure.

Finally, the mathematical modeling was calibrated, considering the irregularities of the ground and the buildings in the mouth area, using Light Detection and Ranging (LIDAR) topography of 1 m $\times$ 1 m pixel. Subsequently, it proceeded to generate the impact susceptibility map considering the worst-case scenario the maximum flow-type landslides for a specific return period. There were included cartography and qualitative classification of the materiality of the buildings present in the area, considering their possible resistance to the displacement of a debris flow. In addition to this, a 3D hydrometeorological model was developed in the ArcScene v10.8 software (ESRI, Redlands, CA, USA). This was with the objective of obtain a closer visualization to the reality about the displacement of the maximum debris flow in the projection of the urban area of La Chimba area, considering the worst scenario.

### 2.2. Validation Analysis

In hyper-arid hydrographic areas where there is no possibility of determining the radius of influence, due to the landslides, by identifying alluvial-type sediments, due to urbanization of the area, and where there is no evidence and/or analysis of the potential impact, developing validation methods to estimate the accuracy of the results of mathematical modeling and obtaining evidence of possible activations of the basin is recommended. This was performed through a geological-sedimentological and morphometric analysis to obtain the hydrological potential and response of the basin. This was complemented with a retrospective flow-type landslides impact zone model through the application of the same methodology in a real basin. The selected basin must have an impact historical, and their rainfalls associated.

### 2.2.1. Soil Analysis

This analysis allows determining the hydrological response due to the geological units and mainly the soil present in the basin. It allows identifying of alluvial sediments, evidencing previous events, and it defines the type of flow most likely to be generated with the associated infiltration capacity, hydraulic conductivity, porosity and other. It is carried out through a geological characterization at a scale of 1:2500 with the purpose of identifying the greatest amount detail. In addition, a granulometric analysis develops through by mechanical sieving of the identified sediment, whose samples were extracted by Electrowatt Engineering [45] (Figure 4).

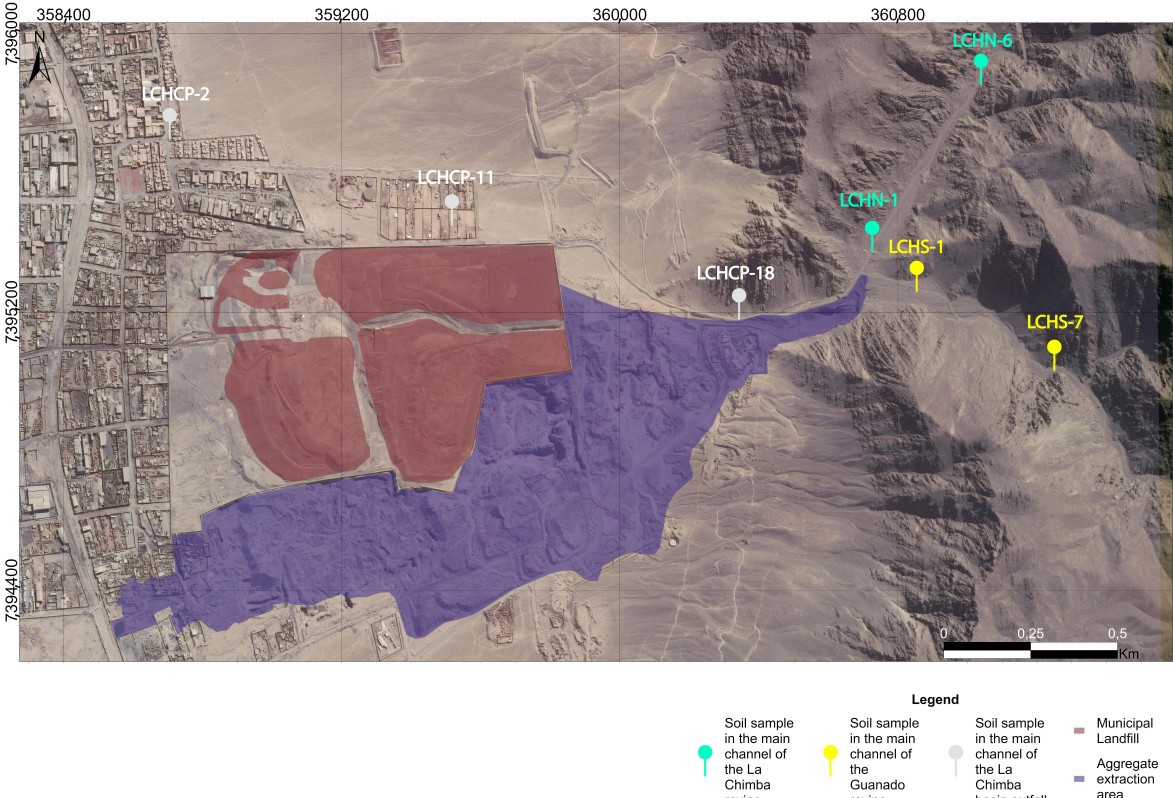

**Figure 4.** Location of pits and soil samples in the study area. The white points and codes indicate samples from the mouth area, light blue points indicate Guanaco sub-basin samples and yellow colors indicate La Chimba sub-basin samples. The brown color represents the Municipality Dump. The purple area represents a sedimentary material extraction zone and ground modification.

The geological characterization it develops at a scale of 1:2500 through the realization of field campaigns to obtain data, samples of geological units, geoprocessing and cartography with the use of ArcGis Pro v2.9.0 software (ESRI, Redlands, CA, USA) with the support of centimetric precision satellite images, Lansat 8, DEM 12.5 m and LIDAR 1 m pixel. In addition, it is complemented with existing bibliographic information on the area.

The granulometric analysis was carried out by representing the results graphically in the "Granulometric Curve" [46]. For its classification, the Unified Soil Classification System (U.S.C.S.) and the sedimentary grain size scale Udden-Wentworth (1922) were used, in addition to the Phi scale of grain sizes ($\Phi$) (Equation (1)).

$$\Phi = -Log_2(D) \tag{1}$$

Subsequently, the statistical parameters were calculated. For this, it was necessary to develop graphs of granulometric curves of frequencies and accumulated frequencies, whose methodology is summarized in Appendix C.

### 2.2.2. Morphometric Analysis

The morphometric analysis, in this case, corresponds to a quantitative analysis of the hydromorphological response of the basin. It was proposed to determine parameters of shape, drainage network, relief and complementary for the definition of hydrologic response of the basin and the ease with which said flow drains towards of the mouth [40,47–51]. Appendix D shows the methodological summary and parameters used in this study. It should be noted that this characterization was carried out using Digital Elevation Models (DEM) of 12.5 m resolution with ArcGIS Pro V2.9.0 geoprocessing and ArcGIS ArcMap Desktop V10.8.2 (ESRI, Redlands, CA, USA).

2.2.3. Basin Modeling

To validate the La Chimba basin mathematical modeling, it replicated the methodology indicated in Section 2.1, except for the calculation of the return period. This was because the modeling was based on the calculation of debris flows for a specific hydrometeorological event, which must have rainfall registers, stream activation and impact zones for subsequent comparison between results. The selection of the hydrographic basin is since it must contain similar characteristics, such as a close location, the existence of mitigation works and flow anthropogenic alteration, such as broken pipes and others. The software used will be the same considered as the modeling selected for the case of La Chimba basin (RAMMS) and the flow density will be fed back in terms of the closest result to the recorded impact zone to be compared.

**3. Results and Discussions**

*3.1. Hydrometeorological Characterization*

3.1.1. Pluviometric Analysis

Based on the results, it is considered that the station of the Dirección General de Aguas (DGA) of the Ministerio de Obras Públicas (MOP), is the one that best represents the historical and current rainfall regime of the study area. This is because it has continuous records, without gaps in them, it is also the station with the least distance from the study area (5.5 km), so the orographic factor and wind direction/intensity are very close to those presented in said area (Appendix E).

3.1.2. Statistical Frequency Analysis

According to the methodology of "Adjustments of Gumbel's probabilistic distribution models". The results are grouped in Tables 2–4.

**Table 2.** Calculation of the probabilities of exceedance, non-exceedance and Return Period (*Tr*) for maximum annual rainfall according to the Gumbel method. A. Years of rainfall: 1978, 1985, 1988, 1990, 1993, 1997, 1998, 1999, 2001, 2003, 2007, 2008, 2010, 2013 and 2018. *Tr*. Return period, which is inversely proportional to the probability of exceedance (*P*(*X* < *x*) %). Source: Data extracted from the Dirección General de Aguas (DGA) [42,52].

| Data Ordered from Highest to Lowest (Rainfall) | | Gumbel Distribution for Extreme Values | | | | |
|---|---|---|---|---|---|---|
| Date | Maximum Daily Rainfall Events (mm) | Gumbel Distribution Probability of Non-Exceedance ($P(X < x)$) | Gumbel Distribution Probability of Non-Exceedance ($P(X < x)$%) | Gumbel Distribution Exceedance Probability ($P(X < x)$) | Gumbel Distribution Exceedance Probability ($P(X < x)$%) | Return Period (*Tr*) (Years) |
| 24 March 2015 | 31.50 | 0.9957 | 99.57% | 0.0043 | 0.43% | 230.1945 |
| 6 June 2017 | 20.50 | 0.9711 | 97.11% | 0.0289 | 2.89% | 34.5495 |
| 17 June 1991 | 17.00 | 0.9475 | 94.75% | 0.0525 | 5.25% | 19.0519 |
| 28 July 1987 | 13.20 | 0.9010 | 90.10% | 0.0990 | 9.90% | 10.1000 |
| 29 August 2006 | 11.50 | 0.8693 | 86.93% | 0.1307 | 13.07% | 7.6525 |
| 24 June 2016 | 6.70 | 0.7246 | 72.46% | 0.2754 | 27.54% | 3.6312 |
| 8 July 2011 | 6.10 | 0.6994 | 69.94% | 0.3006 | 30.06% | 3.3271 |
| 13 January 1983 | 6.00 | 0.6951 | 69.51% | 0.3049 | 30.49% | 3.2795 |
| 27 August 2002 | 3.80 | 0.5869 | 58.69% | 0.4131 | 41.31% | 2.4209 |
| 27 August 1982 | 3.50 | 0.5705 | 57.05% | 0.4295 | 42.95% | 2.3280 |
| 28 May 1992 | 3.00 | 0.5422 | 54.22% | 0.4578 | 45.78% | 2.1841 |
| 7 June 1984 | 2.00 | 0.4828 | 48.28% | 0.5172 | 51.72% | 1.9334 |
| 31 May 2000 | 1.80 | 0.4705 | 47.05% | 0.5295 | 52.95% | 1.8886 |

**Table 2.** *Cont.*

| Data Ordered from Highest to Lowest (Rainfall) | | Gumbel Distribution for Extreme Values | | | | |
|---|---|---|---|---|---|---|
| Date | Maximum Daily Rainfall Events (mm) | Gumbel Distribution Probability of Non-Exceedance ($P(X < x)$) | Gumbel Distribution Probability of Non-Exceedance ($P(X < x)$%) | Gumbel Distribution Exceedance Probability ($P(X < x)$) | Gumbel Distribution Exceedance Probability ($P(X < x)$%) | Return Period ($Tr$) (Years) |
| 20 July 2009 | 1.60 | 0.4581 | 45.81% | 0.5419 | 54.19% | 1.8455 |
| 12 September 2014 | 1.40 | 0.4457 | 44.57% | 0.5543 | 55.43% | 1.8040 |
| 18 May 1986 | 1.00 | 0.4205 | 42.05% | 0.5795 | 57.95% | 1.7257 |
| 20 May 1995 | 1.00 | 0.4205 | 42.05% | 0.5795 | 57.95% | 1.7257 |
| 20 July 1994 | 0.80 | 0.4079 | 40.79% | 0.5921 | 59.21% | 1.6888 |
| 5 August 1981 | 0.70 | 0.4015 | 40.15% | 0.5985 | 59.85% | 1.6709 |
| 26 March 1979 | 0.50 | 0.3888 | 38.88% | 0.6112 | 61.12% | 1.6361 |
| 20 August 1989 | 0.50 | 0.3888 | 38.88% | 0.6112 | 61.12% | 1.6361 |
| 29 August 1996 | 0.50 | 0.3888 | 38.88% | 0.6112 | 61.12% | 1.6361 |
| 27 October 1980 | 0.40 | 0.3824 | 38.24% | 0.6176 | 61.76% | 1.6192 |
| 25 April 2005 | 0.40 | 0.3824 | 38.24% | 0.6176 | 61.76% | 1.6192 |
| 26 July 2004 | 0.10 | 0.3632 | 36.32% | 0.6368 | 63.68% | 1.5705 |
| A | 0.00 | 0.3569 | 35.69% | 0.6431 | 64.31% | 1.5549 |

**Table 3.** Results of the parameters of the Gumbel method equation. Where $\alpha$ and $u$ are parameters of the expression and $\sigma_y$ and $\mu_y$ are values that are based on the number of data or total samples, which are given based on Sánchez [37].

| Number of Data (n) | Mean (Average) ($\dot{X}$) | Trend | Standard Deviation ($S_x$) | $\sigma_y$ | $\mu_y$ | $\alpha$ | $u$ |
|---|---|---|---|---|---|---|---|
| 41 | 3.3 | 0.000 | 6.576 | 1.1413 | 0.5436 | 5.7622 | 0.1726 |

**Table 4.** Rainfall for different specific return periods ($Tr$). $P^{Tr}$ (mm) maximum daily rainfall in a return period of "$x$" years.

| $Tr$ | $1/P^{Tr}$ | $1-1/Tr$ | $P^{Tr}$ (mm) |
|---|---|---|---|
| 2 | 0.5 | 0.5 | 2.284 |
| 5 | 0.2 | 0.8 | 8.815 |
| 10 | 0.1 | 0.9 | 13.140 |
| 20 | 0.05 | 0.95 | 17.287 |
| 50 | 0.02 | 0.98 | 22.656 |
| 100 | 0.01 | 0.99 | 26.679 |
| 200 | 0.005 | 0.995 | 30.688 |

3.1.3. Calculation of Maximum Flow Rates

It is calculated a design rainfall with return period and specific duration ($Pt^{Tr}$) (mm). Tables 5 and 6, and Figure 5 show the values obtained.

**Table 5.** Rainfall for different return periods and different duration. $Pt^{Tr}$ is the rain with a return period of *Tr* years and a duration of *t* hours on a millimeter scale. *K* is the correction coefficient. CDt is the duration coefficient for a period of t hours. CFt is the frequency coefficient for a return period of *Tr* years.

| Duration (*t*) (Hours) | *K* | 1.1 | 1.1 | 1.1 | 1.1 | 1.1 | 1.1 | 1.1 |
| --- | --- | --- | --- | --- | --- | --- | --- | --- |
| | CF | 0.53 | 0.83 | 1 | 1.18 | 1.42 | 1.6 | 1.78 |
| | CDt | Rain $P_t^{Tr}$ (mm) | | | | | | |
| | | $P_t^{Tr}$ **Tr2** | $P_t^{Tr}$ **Tr5** | $P_t^{Tr}$ **Tr10** | $P_t^{Tr}$ **Tr20** | $P_t^{Tr}$ **Tr50** | $P_t^{Tr}$ **Tr100** | $P_t^{Tr}$ **Tr200** |
| 1 | 0.9 | 6.895 | 10.797 | 13.009 | 15.350 | 18.472 | 20.814 | 23.155 |
| 2 | 0.9 | 6.895 | 10.797 | 13.009 | 15.350 | 18.472 | 20.814 | 23.155 |
| 4 | 0.9 | 6.895 | 10.797 | 13.009 | 15.350 | 18.472 | 20.814 | 23.155 |
| 6 | 1 | 7.661 | 11.997 | 14.454 | 17.056 | 20.525 | 23.126 | 25.728 |
| 8 | 1 | 7.661 | 11.997 | 14.454 | 17.056 | 20.525 | 23.126 | 25.728 |
| 10 | 1 | 7.661 | 11.997 | 14.454 | 17.056 | 20.525 | 23.126 | 25.728 |
| 12 | 1 | 7.661 | 11.997 | 14.454 | 17.056 | 20.525 | 23.126 | 25.728 |
| 14 | 1 | 7.661 | 11.997 | 14.454 | 17.056 | 20.525 | 23.126 | 25.728 |
| 18 | 1 | 7.661 | 11.997 | 14.454 | 17.056 | 20.525 | 23.126 | 25.728 |
| 24 | 1 | 7.661 | 11.997 | 14.454 | 17.056 | 20.525 | 23.126 | 25.728 |

**Table 6.** Rainfall intensity (IPtTr) (mm/h) for different return periods of *Tr* years and for different durations of *t* hours.

| Duration (*t*) (Hours) | Rain Intensity (mm/h) | | | | | | |
| --- | --- | --- | --- | --- | --- | --- | --- |
| | $IP_t^{Tr2}$ | $IP_t^{Tr5}$ | $IP_t^{Tr10}$ | $IP_t^{Tr20}$ | $IP_t^{Tr50}$ | $IP_t^{Tr100}$ | $IP_t^{Tr200}$ |
| 1 | 6.895 | 10.797 | 13.009 | 15.350 | 18.472 | 20.814 | 23.155 |
| 2 | 3.447 | 5.399 | 6.504 | 7.675 | 9.236 | 10.407 | 11.578 |
| 4 | 1.724 | 2.699 | 3.252 | 3.838 | 4.618 | 5.203 | 5.789 |
| 6 | 1.277 | 1.999 | 2.409 | 2.843 | 3.421 | 3.854 | 4.288 |
| 8 | 0.958 | 1.500 | 1.807 | 2.132 | 2.566 | 2.891 | 3.216 |
| 10 | 0.766 | 1.200 | 1.445 | 1.706 | 2.052 | 2.313 | 2.573 |
| 12 | 0.638 | 1.000 | 1.205 | 1.421 | 1.710 | 1.927 | 2.144 |
| 14 | 0.547 | 0.857 | 1.032 | 1.218 | 1.466 | 1.652 | 1.838 |
| 18 | 0.426 | 0.666 | 0.803 | 0.948 | 1.140 | 1.285 | 1.429 |
| 24 | 0.319 | 0.500 | 0.602 | 0.711 | 0.855 | 0.964 | 1.072 |

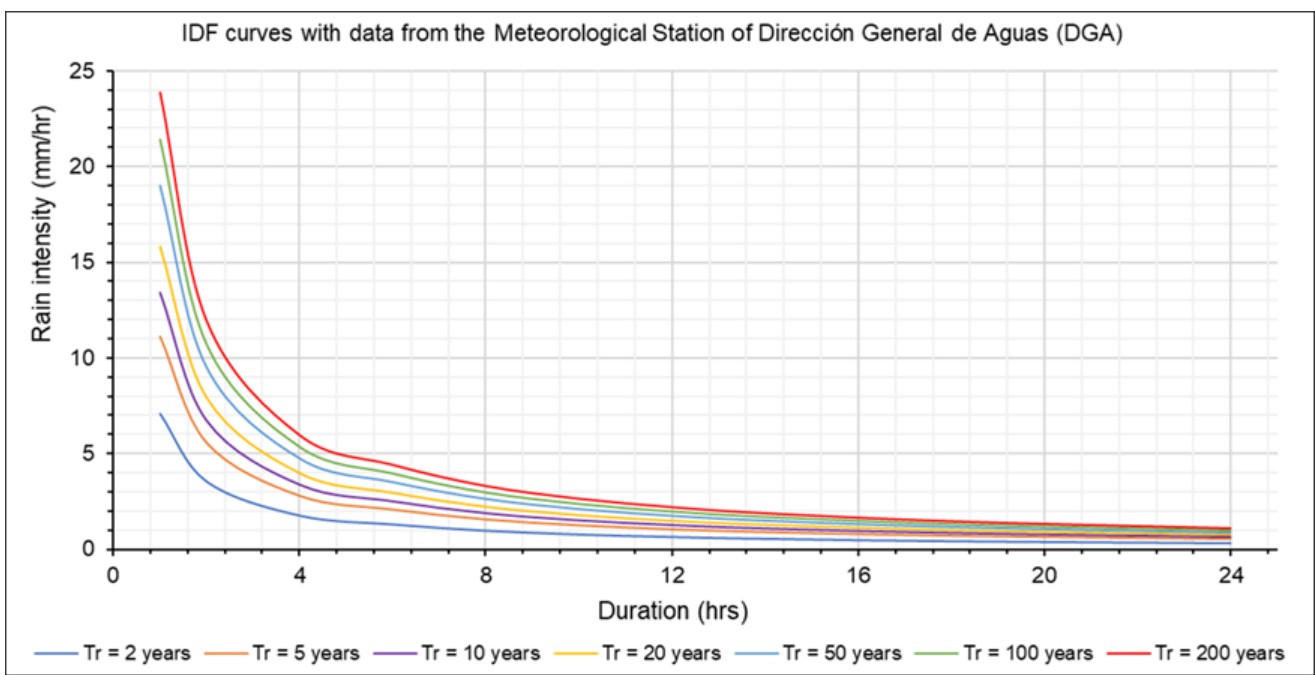

**Figure 5.** Intensity-duration-frequency curves (IDF) based on the record of historical and current rainfall in Antofagasta city, as recommended for this type of analysis [40].

There is constructed the debris flow hydrograph considered for the basin mouth area. For this, the volumetric concentration parameters, concentration times, runoff coefficient, and liquid and debris flows were determined for La Chimba and Guanaco sub-basin, respectively (Table 7).

**Table 7.** Design rainfall intensity (mm/h) and debris flows (m$^3$/s) as a function of concentration time ($T_c$) (h) in the basin.

| | | Result | | |
|---|---|---|---|---|
| Return Period (*Tr*) (Años) | Design Rain Intensity (*i*) (mm/h) | La Chimba Sub-Basin | Guanaco Sub-Basin | Total Detrital Flow ($Q_{DT}$) (m$^3$/s) |
| | | Debris Flow ($Q_D$) (m$^3$/s) | Debris Flow ($Q_D$) (m$^3$/s) | |
| 2 | 7.890 | 17.806 | 8.467 | 26.272 |
| 5 | 12.356 | 27.884 | 13.259 | 41.143 |
| 10 | 14.887 | 33.595 | 15.975 | 49.570 |
| 20 | 17.566 | 43.607 | 20.735 | 64.342 |
| 50 | 21.139 | 57.246 | 27.221 | 84.468 |
| 100 | 23.819 | 67.191 | 31.950 | 99.140 |
| 200 | 26.498 | 74.750 | 35.544 | 110.294 |

Finally, the worst case is debris flows with a return period of 200 years (Tr200) equivalent to 74,750 m$^3$/s for La Chimba sub-basin, 35,544 m$^3$/s for the Guanaco sub-basin and 110,294 m$^3$/s for the area mouth.

### 3.1.4. Flow-Type Landslide Mathematical Modeling

The triangular hydrograph was obtained, designed with a duration of 3 h based on the same duration recorded in the last event of great impact rainfall as in 1991. There are considered Intervals of 5 min, whose debris flow would increase gradually until reaching the maximum flow between 1 h and 25 min and 1 h and 35 min, to then decrease until reaching 0.1 m³/s at 3 h of duration. Therefore, hydrographs were generated for the La Chimba basin and the Guanaco basin (Figure 6). In addition, it is calculated the Manning roughness coefficient (*n*) (see Tables 8 and 9).

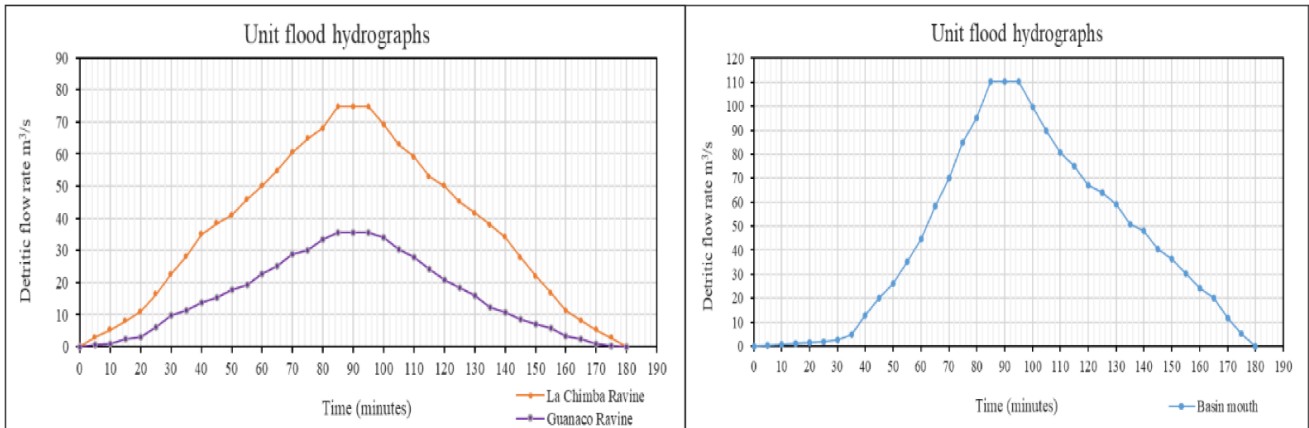

**Figure 6.** Triangular flood hydrograph for La Chimba basin. Represents the debris flow variation rates as a function of the rainfall event duration. The left figure indicates the La Chimba sub-basin (orange line) and the Guanaco sub-basin (purple line) unit hydrograph. The right figure indicates the unit hydrograph of the La Chimba basin mouth.

**Table 8.** Effective diameters for the calculation of $n_o$, which is the Manning coefficient index.

| Data | Required Values | Result |
|---|---|---|
| **Zone** | $D_{90}$ **(mm) (Average)** | $n_o$ |
| La Chimba sub-basin | 0.12 | 0.0266 |
| Guanaco sub-basin | 0.14 | 0.0273 |

**Table 9.** Value of the Manning roughness coefficient sub-indices—the Cowan method. In addition, the value of the volume of the debris flow ($V_d$) is considered with a value of 367,647 m³ and an average velocity ($V_m$) of 3.03 m/s for the maximum intensity of rainfall in a hydrograph time of 1.5 h.

| | Required Values | | | | | | Result |
|---|---|---|---|---|---|---|---|
| **Zone** | **m** | $n_o$ | $n_1$ | $n_2$ | $n_3$ | $n_4$ | $n'$ | **Total** |
| La Chimba sub-basin | 1.15 | 0.0266 | 0.005 | 0.005 | 0 | 0.01 | 0.0535 | Floor; fine-coarse gravel, perimeter irregularity; slight, variation sections; occasionally alternate, obstructions; despicable, vegetation; low and sinuosity; appreciable-high |
| Guanaco sub-basin | 1.15 | 0.0273 | 0.005 | 0.005 | 0 | 0.01 | 0.0543 | Floor; fine-coarse gravel, perimeter irregularity; slight, variation sections; occasionally alternate, obstructions; despicable, vegetation; low and sinuosity; mild |

As a result, the impact zones were determined (Figure 7), which were, then, extrapolated to a three-dimensional model with a detailed mapping of the urban area (Figure 8). Figure 7 allows the comparison between the results delivered by the HEC-RAS and RAMMS software modeling and for maximum, average and minimum fluidity for the last case. It is verified that the delimited area varies considerably depending on the software and the calibration itself, which is key to consider. In the case of HEC-RAS, the modeled flow represents an approximate fluidity of a river flow. On the other hand, RAMMS delivers values where it is possible to vary its fluidity, in addition, its use has become frequent in the Servicio Nacional de Geología y Minería (SERNAGEOMIN) [34] for which the modeling of "High Fluidity" is considered for a period of return of 200 years (Tr200) as the worst-case scenario. However, the "medium" and "minimum fluency" scenarios should not be ruled out as cases that can also develop in the area.

The results show that the impact is concentrated mainly in the front area of the municipal dump, in addition to the northern area, but reaching greater heights in the southeast area, specifically in aggregate extraction. Most of these urban areas susceptible to impacts are of waste material (camping areas) and informal buildings of light material, estimating a low or null resistance to the impact of landslide, with a high probability of triggering a disaster by landslide.

### 3.2. Validation Analysis

#### 3.2.1. Soil Analysis

Geology

This area is composed of sedimentary deposits from the marine and continental environment, volcanic and intrusive rocks. The La Chimba basin is restricted to four units, being from the most to the least abundant: La Negra Formation, Alluvial Deposits, Ancient Alluvial and Colluvial Deposits and Colluvial Deposits spanning from the Jurassic to the Holocene (Figure 9 and Appendix F).

The presence of alluvial deposits in the lower areas of the streams stands out, which effectively show deposits originated by previous hydrometeorological events. In addition, given the low geotechnical quality of the La Negra Formation, it shows a high contribution of colluvial material that increases the volume of available sediments. The La Negra formation, on the other hand, presents the largest extension in the basin, which provides a considerable area of direct runoff causing its direct displacement towards these unconsolidated sedimentary deposits present in the low areas, being able to considerably increase the velocity of the flow, saturation and soil erosion.

Granulometry

There are presented cumulative frequency graphs for the La Chimba and Guanaco sub-basins. In the same way, percentage histograms and frequency curves are made with the associated results (Appendix G).

The final granulometric results of both the effective diameters, percentiles and statistical parameters are attached in Appendix H, where the values obtained are related to their corresponding meaning.

Finally, it was obtained that the basin presents evidence of deposits of alluvial origin with clasts from millimeter to metric, with a soil with an average size of granules to small pebbles, well graded and poorly selected, whose sediment transport current is of low fluidity and of high energy with a medium infiltration coefficient. The hydraulic conductivity coefficient ($K$) showed average values (0.51 to 1.44 cm/h) corresponding to a SM soil based on the Unified Soil Classification System (U.S.C.S.). Regarding the mouth area, it presents low slopes (8% on average) and a notable lateral extension. Its material reflects the type of soil characteristic for this area, in which the material has been deposited in the form of an alluvial fan, being possible to differentiate its proximal, middle and middle-distal sectors by the existence of deposits of conglomerates, sands and sands with higher percentage of fines.

It should be considered that the adopted granulometry methodology does not include clasts larger than 72.2 mm, despite being present in both sub-basins, so it is considered that it fundamentally presents a GM soil (silty gravel, poorly graded mixtures of gravel, sand and silt) (U.S.C.S.), considering that the most likely flow to be generated was a debris flow.

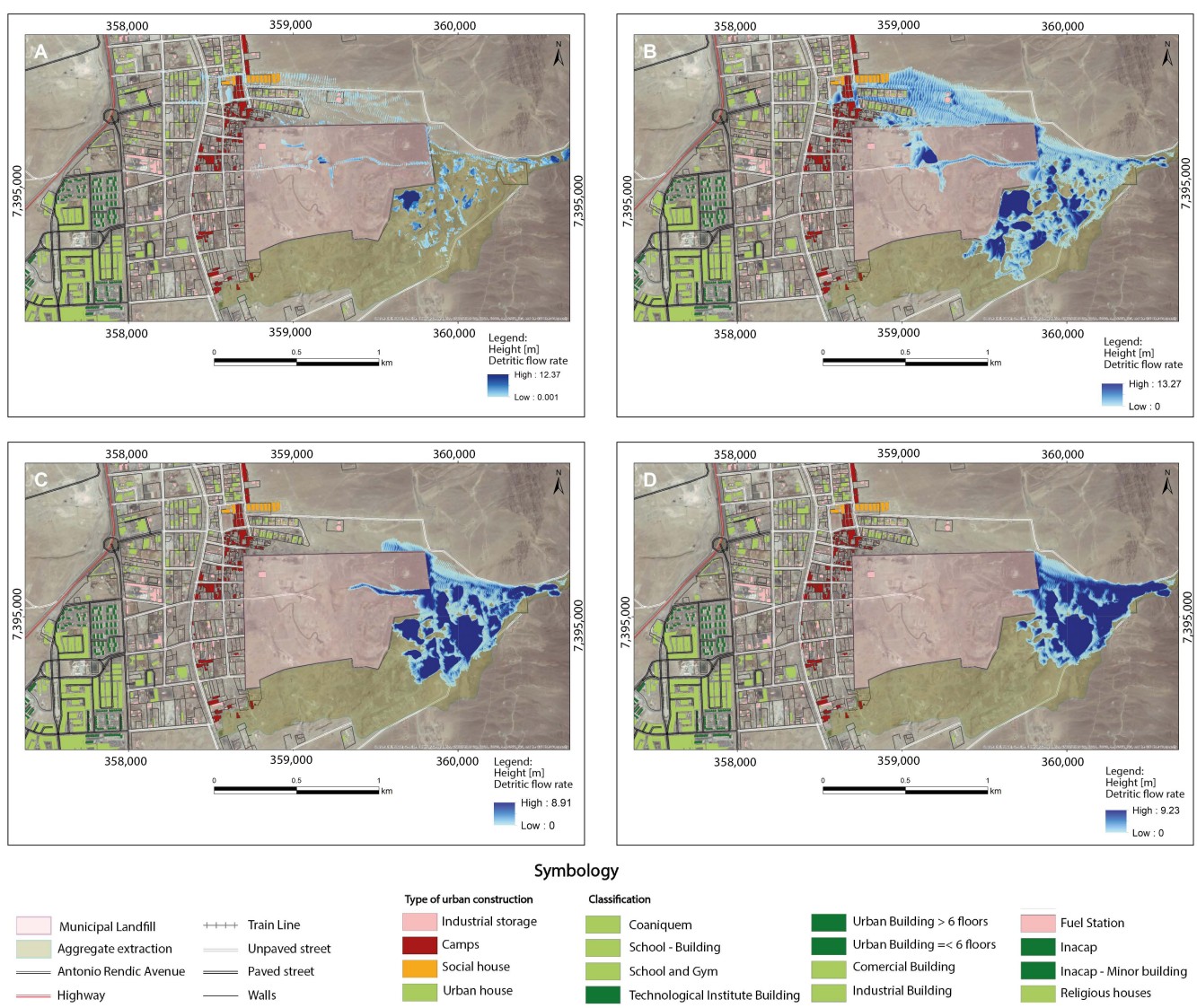

**Figure 7.** Debris flow impact maps for the La Chimba basin. Results of the mathematical modeling of the HEC-RAS software (**A**), RAMMS software for maximum fluidity (**B**), medium fluidity (**C**) and minimum fluidity (**D**) for a maximum debris flow of 110.29 m$^3$ in 1.5 h, representing the maximum intensity of the design event, with a return period of 200 years (Tr200) as the worst scenarios to represent. The red, orange, pink, dark green and light green colors represent from a materiality with null or almost null resistance to a high resistance to landslides such as flows. The bluish colors represent the maximum heights (meters) that debris flows can reach.

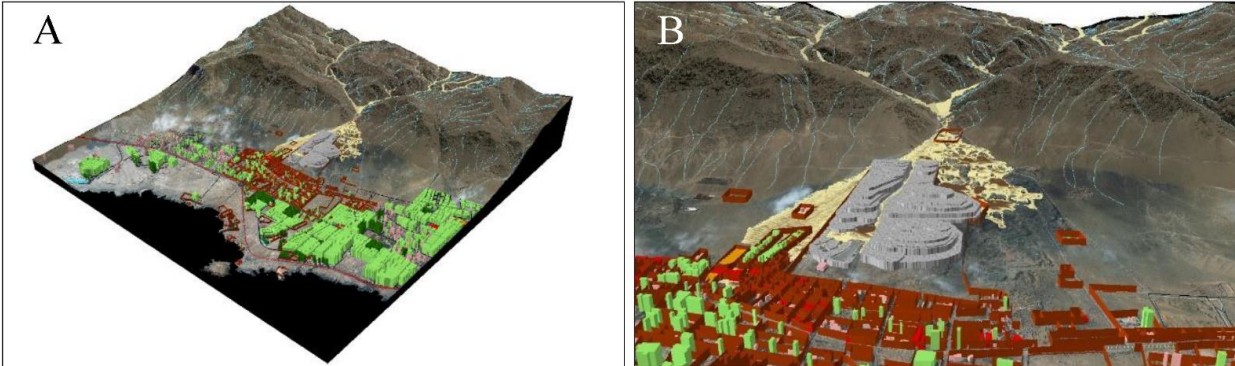

**Figure 8.** Debris flow impact map for the La Chimba basin in 3 dimensions. Results of RAMMS software modeling for maximum fluidity for a maximum debris flow of 110.29 m$^3$ in 1.5 h, representing the maximum intensity of the design rainfall event, with a return period of 200 years (Tr$_{200}$). The red, orange, pink, dark green and light green colors represent from a materiality with null or almost null resistance to a high resistance to landslides such as flows. (**A**) Overview. (**B**) Details of the mouth of the La Chimba basin. The different yellow colorations of the flow indicate the maximum heights of the debris flow.

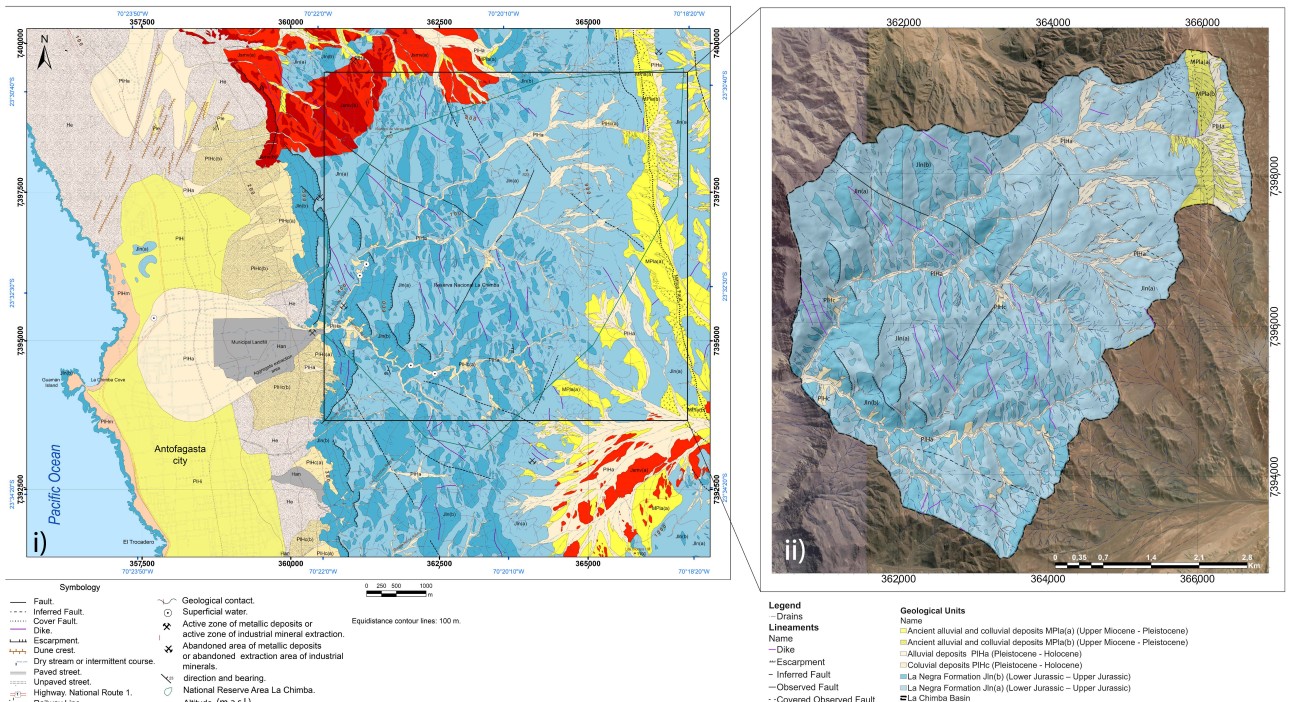

**Figure 9.** Geological units in the northern area of the city of Antofagasta. (**i**) Geological map of the study area. (**ii**) Geological map of La Chimba basin. Han. Anthropic deposits (Holocene). He. Active eolic deposit (Holocene). PlHa. alluvial deposits (Pleistocene–Holocene). PlHc. colluvial deposits (Pleistocene–Holocene) with (b) and without (a) sedimentary intercalations or covers of aeolian sands. PlHi. undifferentiated sedimentary deposits (Pleistocene–Holocene). PlHm. marine deposits (Pleistocene–Holocene). Ple. inactive aeolian deposits (Pleistocene). MPa. old alluvial and colluvial deposits (Upper Miocene–Pleistocene) with (b) and without (a) salt crust cover. Jln. La negra Formation (Lower Jurassic–Upper Jurassic) with (a) and without (b) regolith coverage. Jsmv. quartz diorite and Mantos de Varas tonalite (Upper Miocene–Pliocene) with (a) and without (b) regolith coverage. The delimitation of geological units underlying urban areas is approximate, due to the difficulty of access and visualization. Data from this work and modified from [23,53–55].

3.2.2. Morphometric Analysis

The La Chimba sub-basin and Guanaco sub-basin are considered as a "very small" basin by SEMANART [56] and as "large and intermediate" by Falcón et al. [23], the latter being considered the most appropriate classification for this study, considering the scale and zone on which said classification is based. In turn, the values of La Chimba sub-basin with respect to the perimeter (P), axial length (LA) and an average width (AP) turned out to be greater than the results of the Guanaco sub-basin, which is why it presents a larger catchment area of rainwater. This last idea is reinforced with the calculation of the contributing area since it is 43.5% higher than the Guanaco sub-basin. The Horton Form Factor (F) indicates that the La Chimba sub-basin is an elongated sub-basin, unlike the Guanaco sub-basin which is moderately elongated. The compactness factor ($K_c$), for its part, indicates that the La Chimba sub-basin has an oblong shape, distinct Guanaco, which presents it with an oval tendency, meaning that the latter will have a shorter concentration time than the La Chimba sub-basin (Appendix I).

The parameters of the drainage net (Appendix J), indicate that they are mountainous sub-basins, with intermediate shapes between round and elongated, with typical fluvial systems, in which the geological structures do not influence the shape of the drainage, being generally formed on homogeneous rocks and presenting medium to high concentrations of runoff. In addition, they are well drained, therefore, it indicates that they can present a rapid response to important rainfall events, that is, they would have short concentration times ($T_c$).

The relief parameters (Table 10), indicate that they are "Strongly uneven to uneven". This indicates that it tends to be an area where erosive processes predominate over sedimentation processes, with a tendency to generate flooding in relatively short times, confirming what has been described above.

**Table 10.** Relief parameters, whose value is related to its meaning.

| | Relief Parameters | | | | |
|---|---|---|---|---|---|
| Zone | Maximum and Mínimum Elevation (HM) (Hm) (m a.s.l.) | Absolute Slope (H) (m) | Mean Basin Slope (Pm) (%) | Main Channel Slope (Pcp) (%) | Hypsometric Curve ($C_H$) |
| La Chimba basin | 1062–272 | 790 | 30.285 | 16.200 | Young basin |
| La Chimba sub-basin | 1062–276 | 786 | 26.744 | 16.200 | Young sub-basin |
| Guanaco sub-basin | 988–272 | 716 | 33.825 | 27.558 | Young sub-basin |

This is confirmed by the complementary parameters (Appendix K) that indicate that it is a sub-basin with a high degree of torrentiality. It is interpreted that rainwater, once the rainfall event has started, travels a very short distance to go to the riverbeds, so its discharge speed is greater and, therefore, its torrentiality.

The Hypsometric Curves indicate that both the La Chimba and Guanaco sub-basins present a fluvial morphometry representing a young basin, with unbalanced phases where sediment production and erosive processes predominate, which are in accordance with the geology observed on the ground and represented cartographically in the geological map (Figure 10).

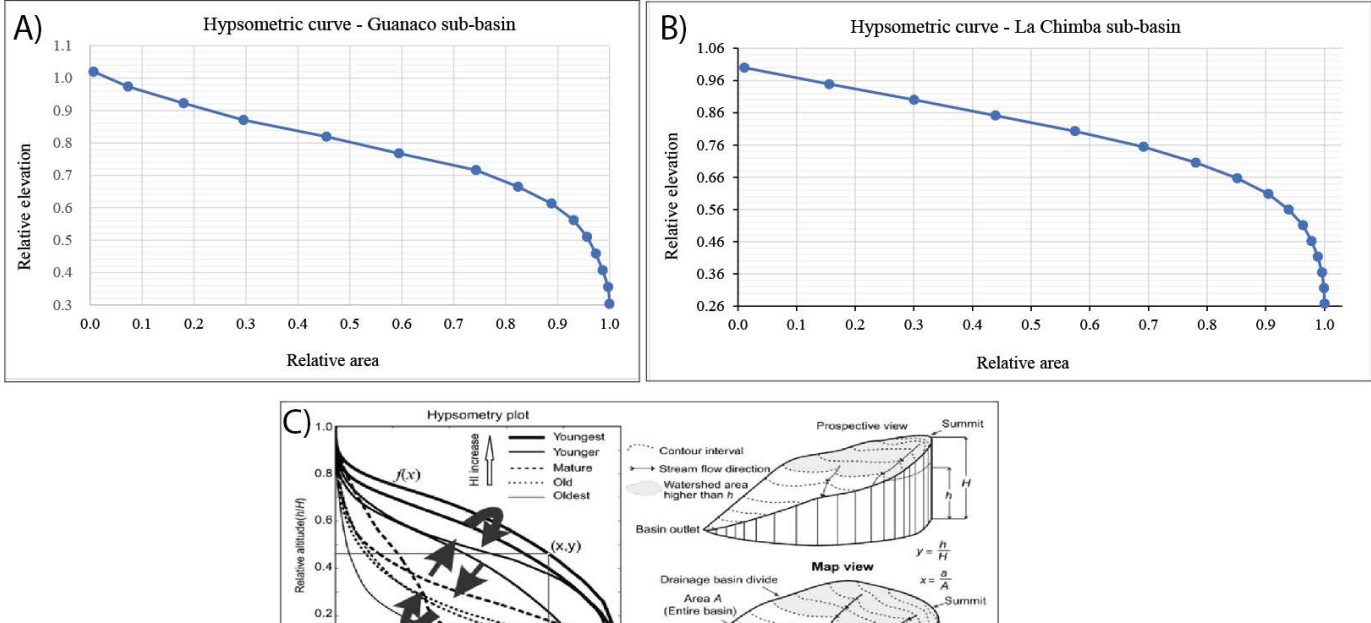

**Figure 10.** La Chimba basin hypsometric curve. (**A**) Guanaco sub-basin hypsometric curve. (**B**) La Chimba sub-basin hypsometric curve. (**C**) The result interpretation is based on a comparative curves resulting from the Strahler [57] methodology, in Liem and Bui [58].

Lastly, the La Chimba basin presents steep slopes, with good drainage, with a tendency to form a runoff in a relatively short time and with insignificant flood maximums. This indicates evidence of an effective response of stream activations to torrential type hydrometeorological events, with a rapid response of the basin to significant rainfall events.

### 3.2.3. Basin Modeling

The results of the validation are presented regarding the comparison between the historical record of the impact zone by flow-type landslides of the 1991 hydrometeorological event (record of Hauser, [28]) and the results of the mathematical modeling in the Riquelme basin (Figure 3) using the same methodology applied in the La Chimba basin (item 2.1). Variations are evident in the south, east and northeast, but with an important trend towards similarity (Figure 11).

The variation of both impact zones may be due to: methodology and implements used in historical cartography, changes in urban areas, standard deviation of the methodology, inaccuracies in the software used, rainfall measurement methodology, among others. However, the results obtained show a tendency towards similarity, so it is estimated that the methodology can effectively be used in basins located in geographical areas ranging from semi-arid to hyper-arid with similar characteristics.

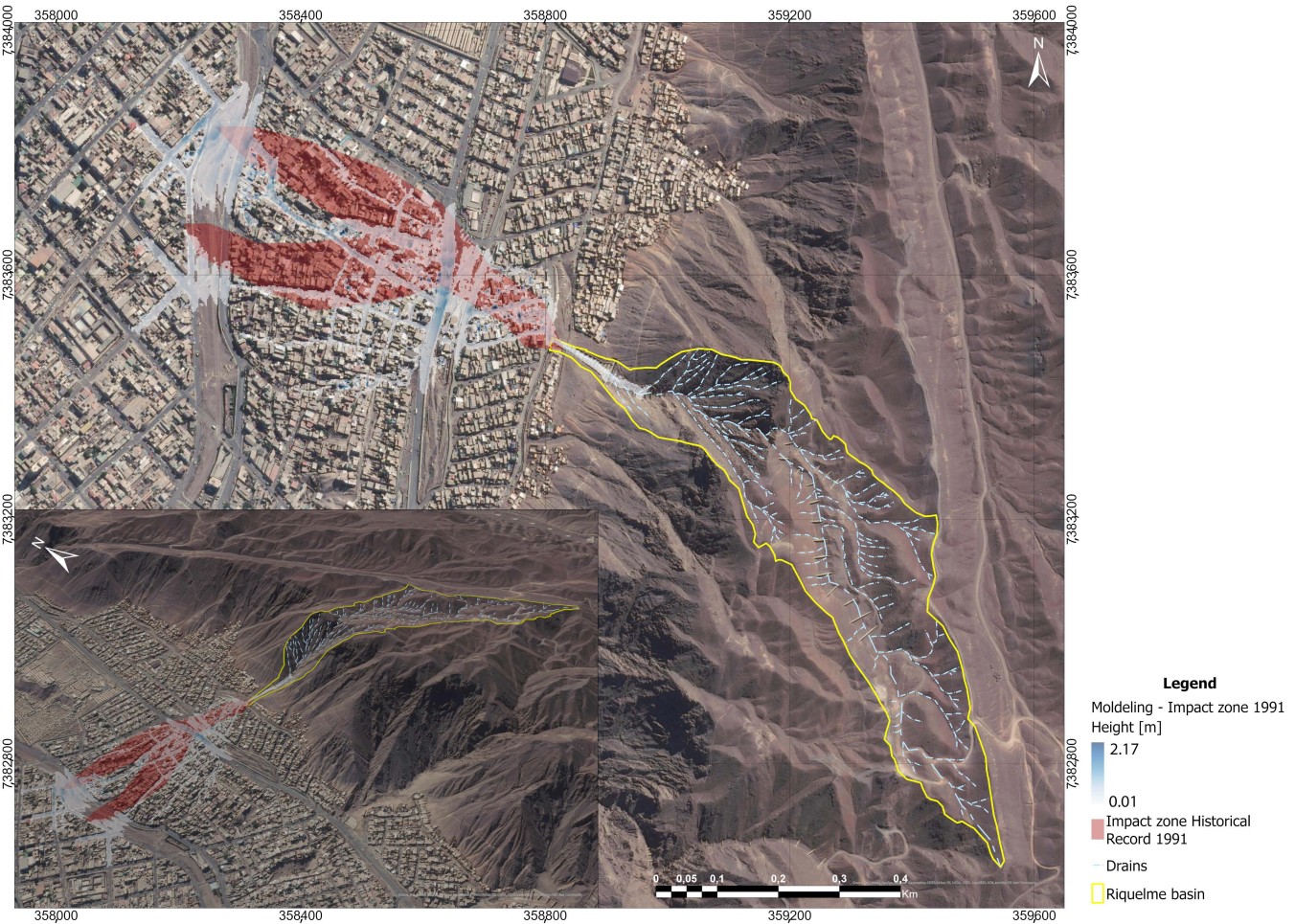

**Figure 11.** 2D and 3D view of the methodological validation map with application in the Riquelme basin between the historical record of the impact zone due to flow-type landslide of the 1991 hydrometeorological event and the mathematical modeling with application of the same methodology of La Chimba basin modeling.

## 4. Conclusions

Drylands are exposed to different types of natural hazards due to their extreme nature and the landslide, but particularly frequent debris flows mainly triggered by heavy and intense rainfall hit these environments and the communities in them. The effects of future climate change on arid regions are poorly known [26]. In the case of Antofagasta city, although it is in the driest desert in the world, the hydrographic basins present evidence of stream activations that have generated flow-type landslides. This shows that the rainfall factor is effective for the development of landslide in mountainous areas from the north of Chile.

The hydrometeorological analysis, together with the validation analyzes (soil and morphometric), even though they were analyzed independently, are interrelated and in turn encompass the main conditioning and triggering factors regarding the generation of flows, being essential the analysis of each one of them in the same study to obtain a good evaluation, validation and approximation of the real characteristics of the basin.

The mathematical modeling of flow-type landslides allows the application of historical rainfall for the realization of methodologies that anticipate possible catastrophes, delimiting in this case, impact zones in areas of extreme aridity where only very punctual and spaced pluviometric events take place. This is fundamental especially in cities in which there is evidence of previous activations, with buildings that are not prepared for the rains, added to the non-existence of alluvial alarm systems with urban areas in the foothills of the Andes. In addition, it is evident that, to carry out these modeling, it is required to; a correct obtaining of the parameters to insert, use of high-resolution topography, availability of LIDAR for the detection of buildings and terrain irregularities and quality and precise pluviometric information, especially considering that it is the base information of this methodology. In addition, to consider the results, a validation stage must be developed that indicates that the basin does indeed present characteristics of previous activations, determination of the basin's hydrological response to possible future events, in addition to modeling validation methods in terms of to the accuracy of the results.

The sedimentological and morphometric characteristics of the La Chimba basin indicate that the type of flow that it can originate would have a rapid response to rainfall events of great intensity or duration, being able to transport clasts of up to 1 m in diameter. Consider that, it is concluded that the modeling provided by HEC-RAS represents a fluvial-type flow. On the other hand, modeling with RAMMS is closer to the consistency of a flow-type landslide, estimating that it is closer to reality. Finally, in the case of the RAMMS software, the modeling of maximum fluency in a period of 200 years (Tr200) is considered like the worst scenario. The results show that the impact is concentrated mainly in the front area of the municipal dump, in addition to the northern area, but reaching greater heights in the southeast area, specifically in aggregate extraction. Most of these urban areas susceptible to impacts are of waste material (camping areas) and informal buildings of light material, estimating a low or null resistance to the impact of landslide, with a high probability of triggering a disaster by landslide.

Finally, La Chimba basin presents geological and geomorphological evidence of previous activations with torrential responses to the development of pluviometric events, whose mathematical modeling yielded high precision, consequently proving that the implemented methodology can effectively be used in basins of hyper-arid zones for the delimitation of areas susceptible to impact by flow-type landslides. In addition, this study shows that despite being in a desert zone, mountainous areas are susceptible to the development of these hazard. Demonstrating at the same time the importance of a correct territorial planning according to the geographical characteristics in which they are located.

**Author Contributions:** Conceptualization, F.R. and I.S.; methodology, F.R. and I.S.; software, F.R.; validation, F.R., G.G. and W.R.; formal analysis, F.R.; investigation, F.R. and I.S.; resources, G.G.; data curation, F.R.; writing—original draft preparation, F.R.; writing—review and editing, F.R., N.T., I.S. and W.R.; visualization, F.R.; supervision, F.R. and G.G.; project administration, F.R.; funding acquisition, I.S. and G.G. All authors have read and agreed to the published version of the manuscript.

**Funding:** This research was funded by Project FONDEF ID 21I10181 and Research Center for Integrated Disaster Risk Management (CIGIDEN) ANID/FONDAP 2011/15110017.

**Institutional Review Board Statement:** Not applicable.

**Informed Consent Statement:** Not applicable.

**Data Availability Statement:** The data presented in this article are publicly available on request.

**Acknowledgments:** I.S., N.T., W.R. and F.R. are grateful for the financial support from the Project FONDEF ID 21I10181. F.R. and G.G. thanks for the financial support to Research Center for Integrated Disaster Risk Management (CIGIDEN) ANID/FONDAP 2011/15110017.

**Conflicts of Interest:** The authors declare no conflict of interest.

## Appendix A. Methodological Summary with Its Equations Used, Corresponding to the Hydrometeorological Analysis for the Calculation of Maximum Liquid and Debris Flows for La Chimba Basin, Antofagasta (Chile)

| Hydrometeorological Characterization | | | |
|---|---|---|---|
| **Analysis** | **Method** | **Method—Equation** | **Description** |
| Statistical frequency analysis | Probabilities of non-excess | $P(X < x) = e^{-e^{\frac{-(x-u)}{\alpha}}}$ | $P(X < x)$ is the probability of non-excess, $P(X > x)$ is the probability of excess, $e$ is the natural number, $Tr$ is the return period, $x$ are the recorded rainfall events and $\alpha$ and $u$ are eigenmetric parameters of the expression. $\dot{x}$ is the arithmetic mean of the samples, $S_x$ is the standard deviation of the samples and $\sigma_y$ and $\mu_y$ are values that are a function of the total number of total samples [37–39]. |
| | Probabilities of excess | $P(X > x) = 1 - e^{-e^{\frac{(x-u)}{\alpha}}} = \frac{1}{Tr}$ | |
| | Parameters of the expression | $\alpha = \frac{S_x}{\sigma_y}$ | |
| | Rainfall for a specific return period (mm) | $u = \dot{x} - \mu_y \times \alpha$ $P^{Tr} = -Ln\left(-Ln\left(1 - \frac{1}{Tr}\right)\right) \times \alpha + \mu$ | $P^{Tr}$ is the rainfall event recorded (mm) for a specific return period ($Tr$), Ln is the natural logarithm and $\alpha$—$\mu$ are parameters of the equation described above [37]. |
| Calculation of maximum flows | IDF Curves | $P_t{}^{Tr} = K \times CD_t \times CF_T \times P^{Tr10}$ | $Pt^{Tr}$ is the rain with return period $Tr$ years and duration of $t$ hours, $P^{Tr10}$ is the maximum daily rainfall (24 h) with 10 years of return period, $CDt$ is the duration coefficient for $t$ hours, CFT is the frequency coefficient for $T$ years of return period and $K$ is the correction coefficient for the maximum rainfall $P^{Tr10}$ measured between 8 am and 8 am with respect to the 24 rainiest hours of the storm, which, according to the Manual of Roads [38], adopts a value of 1.1. CFT is the frequency coefficient for a return period $T$. $Pt^{Tr}$ is the rainfall of duration $t$ and return period of $T$ years. $Pt^{Tr10}$ is rainfall of duration $t$ and return period of 10 years. The duration coefficients and frequency coefficients for cities in Chile with the same return period were extracted from DICTUC [59]. |
| | Frequency coefficient for a return period ($CD_t$) | $CD_t = \frac{P_t{}^{Tr10}}{P_{24}{}^{Tr10}}$ | |
| | Design rainfall intensity for a return period $Tr$ (años) | $IP_t{}^{Tr} = \frac{P_t{}^{Tr}}{t}$ | $IP_t{}^{Tr}$ is the intensity of rainfall for a return period of $Tr$ years and for a duration of t hours, $IP_t{}^{Tr}$ is the rainfall for a return period of $Tr$ years and for a duration of $t$ hours and $t$ is the duration in hours. From these data we proceed to graph the IDF curve [59]. |
| | Rational Method. Maximum liquid flow rate $Q_L$ (m³/s) | $Q_L = \frac{C \times i \times A_a}{3.6}$ | Calculation by generating a hydrogram from the rational method and empirical expression. The calculation of several parameters is required including the runoff coefficient ($C$) (dimensionless, where $0 \leq C \leq 1$), which includes basin characteristics such as: relief, infiltration, vegetation cover and surface storage [40]. |
| | Concentration time $T_c$ (minutos) | $T_c = 0.95 \times \left(\frac{(L_{cp})^3}{H}\right)^{0.385}$ | Expression of California Highways [41], ideal for mountain basins and arid character, as in the case under study, in which $T_c$ is the concentration time (hours), $L_{cp}$ is the length of the main channel (km) and H corresponds to the unevenness from the starting point of the channel to the mouth point (m). |
| | O'Brien and Julien Method (1997) [60] Detritic Maximum Flow Rate ($Q_D$) | $Q_D = Q_L \times BF = Q_L \times \frac{1}{1-C_v}$ | $Q_D$ is the maximum detrital flow rate (m³/s), $Q_L$ is the liquid runoff flow (m³/s) and BF is the factor called bulking factor, where $C_v$ is the volumetric concentration of solids, whose value is extracted from the study of Ayala et al. [42] |

## Appendix B. Some Outstanding Equations that Mathematically Govern the HEC-RAS and RAMMS Software, in Addition to the Equations for Calculating the Parameters Required to Be Inserted

| Hydrometeorological Characterization | | | |
|---|---|---|---|
| **Software** | **Method** | **Method—Equation** | **Description** |
| HEC-RAS v6.1 | Flow heights in a non-permanent flow regime | $Q_D = \left(\frac{A_h}{n_D}\right) \times R_h{}^{\frac{2}{3}} \times S_f{}^{\frac{1}{2}}$ | $Q_D$ (m³/s) corresponds to the detrital flow, $A_h$ (m²) is the wet area or cross-sectional area of the detrital flow, $R_h$ (m) is the hydraulic radius associated with the different cross-sections considered, respectively, $n_D$ corresponds to the Manning coefficient for a detrital flow which depends on the roughness of the walls and $S_f$ represents the slope of the potential line [61]. |
| | Cowan Methodology— Manning Roughness Coefficient ($n_D$) | $n_D = (n_b + n_1 + n_2 + n_3 + n_4) \times m$ | $n_b$ describes the quality of the material present in the floor of the ravine, $n_1$ considers the effect of superficial irregularities, $n_2$ quantifies the variations in shape and size of the different cross-sections, $n_3$ is the relative effect of obstructions in the channel, $n_4$ quantifies the density of vegetation and m is a corrective factor that considers the degree of sinuosity or meander of the channel. For the case study, $n_b$ was determined by granulometric data from soil samples from the respective streams [62]. |

| Hydrometeorological Characterization | | | |
|---|---|---|---|
| **Software** | **Method** | **Method—Equation** | **Description** |
| RAMMS v1.7.20 | Total friction ($S_f$) | $S_{fx} = n_{U_x}\left[\mu g_Z H + \frac{g\|U\|^2}{\xi}\right]$ <br><br> $S_{fy} = n_{U_y}\left[\mu g_Z H + \frac{g\|U\|^2}{\xi}\right]$ | Flow height $H(x, y, t)$ (m) and mean velocity $U(x, y, t)$ (m/s). In turn, consider a total friction ($S_f$) in the following equation. $n_{U_x}$ and $n_{U_y}$ are the directional unit velocity vectors in the x and y directions, respectively. The total basal friction in the Voellmy–Salm model is divided into a velocity-independent Coulomb dry friction coefficient $\mu$ ($M_u$) and a velocity-dependent turbulent friction coefficient $\xi$ ($X_i$) (m/s$^2$) [63–65]. |
| | Takahashi Method (1991)—Volume of solids in the flow ($V_d$) | $V_d = 1500 \times \left[\frac{Q_D}{1-C_V}\right]$ | $V_d$ is the Volume of solids in the flow (m$^3$), $Q_D$ is the maximum detrital flow (m$^3$/s) and $C_V$ corresponds to the volumetric concentration (-). |
| | Mean detrital flow rate ($V_m$) (m/s) | $Vm = \frac{1}{n_D} \times R^{\frac{2}{3}} \times i^{\frac{1}{2}}$ | V(h) is the average flow rate in m/s, as a function of the water height h, $R$ corresponds to the hydraulic radius of the section considered (m), n represents the Manning roughness coefficient and $i$ is the slope of the water line (m/m) [40]. |

## Appendix C. Statistical Parameters Used in the Granulometric Analysis of La Chimba Basin

| Name | Method—Equation | Description |
|---|---|---|
| Gradation of the soil. Coefficient of uniformity ($C_u$) and Coefficient of curvature ($C_c$) [66] | $C_u = \frac{D_{60}}{D_{10}}$ <br><br> $C_c = \frac{(D_{30})^2}{D_{10} \times D_{60}}$ | The uniformity coefficient ($C_u$) is the extension of the granulometric distribution curve. The coefficient of curvature ($C_c$) gives information regarding the distribution of intermediate sizes. Both coefficients are used as criteria in the Unified Soil Classification System (U.S.C.S.) |
| Hydraulic conductivity coefficient ($K$) [67] | $K = \left(8.64 \times Ca \times ((D10)^2)\right) \div 24$ | $K$ is the hydraulic conductivity (cm/h), D10 is the effective size of the sediments (mm) (10% smaller and 90% larger) and $Ca$ is a coefficient that depends on grain size and uniformity [39]. The effective diameter is calculated directly from the cumulative frequency graph. The factor of 8.64 allows us to enter the value of D10 in mm and we obtain the result of $K$ in m/day. |
| Average Graphic Size ($Mz$) [68] | $Mz = \frac{(\varnothing16 + \varnothing50 + \varnothing75)}{3}$ | It corresponds to the measure of the mean size of the sample in phi units ($\Phi$). Mz corresponds to the mean size of the graph on a phi scale ($\Phi$) and $\Phi16$, $\Phi50$, $\Phi75$ and $\Phi5$ correspond to percentiles with their corresponding percentage. The final result will be evaluated according to the Udden-Wentworth classification and will be indicative of the average kinetic energy of the current. |
| Inclusive graphical standard deviation ($\Phi i$) [68] | $\varnothing i = \frac{\varnothing84 - \varnothing16}{4} + \frac{\varnothing95 - \varnothing5}{6.6}$ | A measure of spread, indicating how far values may be from the average (mean). $\Phi i$ is the inclusive standard deviation, $\Phi84$, $\Phi16$, $\Phi95$ and $\Phi5$ are percentiles with their corresponding percentage. It provides information on the level of selection of the sample and therefore it is a very sensitive index to define the fluidity of the transport and sedimentation medium. |
| Degree of inclusive graphic skewness ($Ski$) [68] | $Ski = \frac{\varnothing16 + \varnothing84 - (2 \times \varnothing50)}{2 \times (\varnothing84 - \varnothing16)} + \frac{\varnothing5 + \varnothing95 - (2 \times \varnothing50)}{2 \times (\varnothing95 - \varnothing5)}$ | Ski is the degree of inclusive graphic skewness, $\Phi5$, $\Phi16$, $\Phi50$, $\Phi84$ and $\Phi95$ are percentiles with their corresponding percentage. |
| Measurement of graphic Kurtosis ($K_G$) [68] | $K_G = \frac{\varnothing95 - \varnothing5}{2.44 \times (\varnothing75 - \varnothing25)}$ | Many curves designated as "normal" by the skewness measure are markedly abnormal when calculated by kurtosis. If the central part of the curve has better selection than the extremes, the curve is leptokurtic, while if the selection is better at the extremes, the curve is platychortic. |
| Mode [46] | - | There may be one or more modes giving rise to unimodal (one), bimodal (two) or multimodal (greater than two) distributions, respectively. In the latter case, the most abundant is called the main mode and the other modes are secondary. |

## Appendix D. Methodological Summary Corresponding to the Morphometric Analysis of La Chimba Basin, Antofagasta (Chile)

| Form Parameters | | |
|---|---|---|
| **Name** | **Equation or Method** | **Description** |
| Basin Area (*A*) [km$^2$] | | A measure of the surface area of a basin, defined as the orthogonal projection of the entire drainage area of a runoff system flowing directly or indirectly into the basin [56] |
| Basin Perimeter (*P*) [km] | | It is defined as the measurement of the watershed envelope line, by the topographic watershed [51] |
| Axial Length (*A$_l$*) [km] | Geographic information system (GIS) | Distance in a straight line between the mouth and the farthest point on the perimeter (*P*) of the basin, which in some cases coincides with the length of the main course [51] |
| Length of the main channel (*Lc*) [km] | | Represents the length of the channel over its entire course (km), including all the sinuosity of the channel. |
| Form Factor (*F*) [69] | $F = \frac{A \ (km^2)}{(Lc)^2 \ (km)}$ | It is defined as the ratio between the area (*A*) and the length of the drainage basin (*L$_c$*). |
| Compactness Factor (*K$_c$*) [70] | $K_c = 0.28 \times \left( \frac{P \ (km)}{\sqrt{A \ (km^2)}} \right)$ | This factor is the oldest one, expressing the relationship between the perimeter of the drainage basin and that of a circle of equal area (equivalent circle); thus, the higher the coefficient, the more distant the shape of the basin will be with respect to the circle. *P* represents the perimeter (km) and A the area (km$^2$) of the Macul basin. |
| **Drainage System Parameters** | | |
| **Name** | **Equation or method** | **Description** |
| Drainage order (*n*) [57] | Geographic information system (GIS) | Horton [69] suggests a hierarchization of streams according to order number as a measure of the branching of the main channel in a basin. This system is dimensionless and was later improved and slightly modified by Strahler [57], indicating that a stream may have one or more segments. |
| Bifurcation ratio (*B$_r$*) [57] | $B_r = \frac{n_i}{n_{i+1}}$ | It is the ratio between the total number of drains of a certain order (*n$_i$*) and the total number of drains of the next higher order (*n$_{i+1}$*). |
| Length Ratio (*L$_r$*) [57] | $L_r = \frac{L_i \ (km)}{L_{i-1}(km)}$ | The ratio of the average length of a certain order of drainage (*L$_i$*) of the average length of the order of drainages that is immediately lower (*L$_{i-1}$*). |
| Drainage network density (*Dd*) [71] [1/km] | $Dd = \frac{\Sigma L_i \ (km)}{A \ (km^2)}$ | Quotient between the total length of the channels of all of the orders that make up the river system of the basin ($\sum L_i$) and the total area of the basin (*A*). |
| Drainage Frequency (*F*) [72] [1/km$^2$]. | $F = \frac{n_t}{A \ (km^2)}$ | It is defined as the quotient between the total number of river courses (*n$_t$*) and the area of the basin (km$^2$). When obtained, it quantifies the potential for any drop of water to find a channel in within an arbitrary timeframe. |
| Drainage hierarchy (J) | Geographic information system (GIS) | Represents the highest drainage order, obtained using Strahler's [57] drainage order methodology. |
| **Relief Parameters** | | |
| **Name** | **Equation or method** | **Description** |
| Absolute elevation difference (*H*) [ m a.s.l.] | $H = (HM - Hm)$ | Corresponds to the difference between the maximum elevation (*HM*) and the minimum elevation (*Hm*), measured in meters above sea level (m a.s.l.). |
| Average slope of the basin (Sm) [%] | Geographic information system (GIS) | The average slope of a watershed is directly related to the degradation process to which a watershed is subjected [73]. |
| Hypsometric curve [57] | Geographic information system (GIS) and mathematical calculations by calculating relative elevation and relative area, and then applying the results to a graph | The hypsometric curve suggested by Langbein et al. [74] graphically represents the elevations of the terrain as a function of the corresponding surfaces. According to Strahler [57], the importance of this relationship lies in the fact that it is an indicator of the state of dynamic equilibrium of the basin, so the basin can be in a state of youth (disequilibrium), in a state of maturity (equilibrium) or at intermediate levels. |

| Complementary Parameters | | |
|---|---|---|
| Name | Equation or method | Description |
| Torrentiality coefficient ($T_c$) [73] (1/km²) | $T_c = \frac{n_1}{A\ (km^2)}$ | Index that measures the degree of torrentiality of the basin, by means of the ratio of the number of drainages of order 1 ($n_1$) with respect to the total area of the basin ($A$). |
| Potentiality index ($P_i$) [75] | $P_i = \frac{\left(Dd\ \left(\frac{1}{km}\right)+F\ \left(\frac{1}{km^2}\right)+J\right)}{A\ (km^2)}$ | It determines the location of erosion and accumulation zones in a watershed; its determination is important. A high $P_i$ value will reveal that in a specific hydrological basin there is accumulation of debris, which could be transported if high rainfall occurs, as to generate an alluvial event [76] |

## Appendix E. Base Data and Comparative Data from Meteorological Stations with Available Records for the City of Antofagasta

| Institution | Name Estation | Coordinates | Measurement Height (m a.s.l.) | Range of Years with Data Availability | Rainfall Record—Orographic Effect on the Records—Event 1991 | Distance in Relation to the Study Area (km) | Differences between the Average Maximum Rainfall Measurements in 24 h per Year between 1968 and 2018 |
|---|---|---|---|---|---|---|---|
| (DMC) | Portezuelo | 23°42′ S 70°24′ W | 550 | 1904–1944 (40 years) | 14.1 | 14.2 | 1.53 (DMC/UCN-DGA) |
| | Cerro Moreno | 23°27′ S 70°26′ W | 119–137 | 1946–2018 (72 years) | | | |
| (DMC) | Universidad Católica del Norte (UCN) | 23°41′ S 70°25′ W | 30 | 1968–2018 (50 years) | 42.0 | 17.5 | 1.56 (DGA—DMC/ Cerro Moreno) |
| (DGA) | DGA | 23°35′ S 70°23′ W | 50 | 1978–2018 (40 years) | 17.0 | 5.5 | 2.17 (DMC/ UCN—DMC/ Cerro Moreno) |

Notes: DMC. Chilean Meteorological Directorate. DGA. General Directorate of Waters.

## Appendix F. Area and Description of the Geological Units Present in the La Chimba Basin

| Name | Area [km²] | Description |
|---|---|---|
| La Negra Formation Jln (a) (Lower Jurassic–Upper Jurassic) | 15,725 | Andesitic lavas and pyroxene andesites of gray to greenish gray colors with aphanitic, porphydic, brechosal and tonsilloidal textures with subordinate levels of sedimentary gaps and medium grain sandstones. It emerges as a continuous strip of direction NNE—SSW forming a monoclinal sequence with general attitude of N5°–10° W/55°–70° W, which reaches a minimum potential of 5050 m. |
| La Negra Formation Jln (b) (Lower Jurassic–Upper Jurassic) | 5902 | |
| Alluvial deposits PlHa (Pleistocene–Holocene) | 2111 | Gravels and sands unconsolidated to slightly cemented that make up the filling of the active ravines and the alluvial fans of the Cordillera de la Costa. The gravels are clastosoportadas with poorly graded clasts and the matrix consists mainly of coarse sands to silts. They present horizontal and locally paleochannel stratification, grain-decreasing tendencies and imbrications. |
| Ancient alluvial and colluvial deposits MPla (b) (Upper Miocene–Pleistocene) | 0.404 | Unconsolidated to semi-consolidated gravels and sands, distributed in the eastern sector of the study area including the La Chimba basin. They represent continental deposits of piedmont and mud flows originated by gravitational flows and sporadic water contributions under a desert climate where they form cones of medium to strong slope. |
| Ancient alluvial and colluvial deposits MPla (a) (Upper Miocene–Pleistocene) | 0.327 | |
| Coluvial deposits PlHc (Pleistocene–Holocene) | 0.242 | Poorly stratified gravels and sands, unconsolidated to moderately cemented, in centimeter to metric layers distributed on the slopes of steep slopes of the Costa Mountain Range. The gravels are clastosoportadas to matrix supported. The clasts have a poor selection, they are angular with low sphericity and the matrix are fine gravels and coarse sands of grayish brown tones. The sands are coarse in size with regular to a good selection. |

Notes: Source: Data from this study, supplemented with information from [23,53–55].

## Appendix G. Granulometric Results for the La Chimba and Guanaco Sub-Basins. The Values in Gray Are Reference Values since They Were Not Used to Calculate the Parameter in Question

| | La Chimba Sub-Basin | | | | | | |
|---|---|---|---|---|---|---|---|
| **Samples Codes →** | **LCHN-1** | | | **LCHN-6** | | | |
| **Statistical Parameters** | **Opening (mm)** | **Aperture** | **Result** | **Opening (mm)** | **Aperture** | **Result** | |
| | | **Phi ($\Phi$)** | | | **Phi ($\Phi$)** | | |
| Coefficient of uniformity ($C_u$) | 29.167 | −4.866 | Very well graded | 27.500 | −4.781 | Very well graded | |
| Coefficient of curvature ($C_c$) | 1.339 | −0.421 | Well graded | 1.237 | −0.307 | Well graded | |
| Hydraulic Conductivity Coefficient ($K$) [cm/h] | 41.472 | | SP-SW (U.S.C.S) | 41.472 | | SP-SW (U.S.C.S) | |
| | (cm/h) | | | (cm/h) | | | |
| Average Chart Size ($Mz$) | 2.378 | −1.250 | Granule, flow and average energy of the current | 2.351 | −1.233 | Granule, flow and average energy of the current | |
| Inclusive graph standard deviation ($\Phi i$) | 0.149 | 2.748 | Very poorly selected. Low fluidity and high energy current | 0.149 | 2.748 | Very poorly selected. Low fluidity and high energy current | |
| Degree of inclusive graphic asymmetry ($Ski$) | 0.892 | 0.165 | Asymmetrical towards fine | 0.9011 | 0.150 | Asymmetrical towards fine | |
| Measurement of graphical Kurtosis ($KG$) | 0.520 | 0.943 | Mesokurtic | 0.501 | 0.998 | Mesokurtic | |
| Mode | | −1; 3.6 | Bimodal: Granules—fine sands | | −1; 3.6 | Bimodal: Granules—fine sands | |
| Unified Soil Classification System (U.S.C.S.) | SW | | Well-graduated sands, sands with gravel, with few or no fines | SW | | Well-graduated sands, sands with gravel, with few or no fines | |
| | SM | | Silty sands, poorly graded sand and silt mixtures | SM | | Silty sands, poorly graded sand and silt mixtures | |
| | Guanaco Sub-basin | | | | | | |
| **Samples codes →** | **LCHS-1** | | | **LCHS-7** | | | |
| Coefficient of uniformity ($C_u$) | 25.000 | −4.644 | Very well graded | 33.929 | −5.084 | Very well graded | |
| Coefficient of curvature ($C_c$) | 1.210 | −0.275 | well graded | 2.350 | −1.232 | well graded | |
| Hydraulic Conductivity Coefficient ($K$) [cm/hr] | 115.2 | | SP-SW (U.S.C.S) | 56.448 | | SP-SW (U.S.C.S) | |
| | (cm/h) | | | (cm/h) | | | |
| Average Chart Size ($Mz$) | 3.523 | −1.817 | Small pebble, flow and average energy of the current | 3.287 | −1.717 | Small pebble, flow and average energy of the current | |
| Inclusive graph standard deviation ($\Phi i$) | 0.150 | 2.736 | Very poorly selected. Low fluidity and high energy current | 0.169 | 2.563 | Very poorly selected. Low fluidity and high energy current | |
| Degree of inclusive graphic asymmetry ($Ski$) | 1.060 | −0.084 | Almost asymmetrical | 0.802377 | 0.318 | Very asymmetrical towards fine | |
| Measurement of graphical Kurtosis ($KG$) | 0.484 | 1.045 | mMesokurtic | 0.453 | 1.142 | Leptokurtic | |
| Mode | - | −4.6; 0.2; 3.6 | Multimodal: guijarro pequeño, arena muy gruesa y arena fina | - | −1; 3.6 | Bimodal: granules and fine sand | |
| Unified Soil Classification System (U.S.C.S.) | SW | | Well-graduated sands, sands with gravel, with few or no fines | SW | | Well-graduated sands, sands with gravel, with few or no fines | |
| | SM | | Silty sands, poorly graded sand and silt mixtures | SM | | Silty sands, poorly graded sand and silt mixtures | |

## Appendix H. Granulometric Results for the Mouth Area. The Values in Gray Are Reference Values Since They Were Not Used to Calculate the Parameter in Question

| Basin Mouth | | | |
|---|---|---|---|
| **Samples Code** | | LCHCP-2 | |
| **Statistical Parameters** | **Opening (mm)** | **Aperture** | **Result** |
| | | **Phi ($\Phi$)** | |
| Coefficient of uniformity ($C_u$) | 25.000 | −4.644 | Very well graded |
| Coefficient of curvature ($C_c$) | 1.210 | −0.275 | Well gradado |
| Hydraulic Conductivity Coefficient ($K$) [cm/hr] | 165.888 | | SP-SW (U.S.C.S) |
| | (cm/hr) | | |
| Average Chart Size ($Mz$) | 3.647 | −1.867 | Small pebble, flow and average current energy |
| Inclusive graph standard deviation ($\Phi i$) | 0.223 | 2.164 | Very poorly selected. Low fluidity and high energy current |
| Degree of inclusive graphic asymmetry ($Ski$) | 1.226 | −0.294 | Asymmetrical towards thick |
| Measurement of graphical Kurtosis ($KG$) | 0.653 | 0.615 | Very platykurtic |
| Mode | 25.4, −0.84, −0.08 | - | Multimodal: small pebble, granules and fine sand |
| Unified Soil Classification System (U.S.C.S.) | SW | | Well-graduated sands, sands with gravel, with few or no fines |
| | SM | | Silty sands, poorly graded sand and silt mixtures |
| **Samples code** | | LCHCP-11 | |
| **Statistical parameters** | **Opening (mm)** | **Aperture** | **Result** |
| | | **Phi ($\Phi$)** | |
| Coefficient of uniformity ($C_u$) | 36.364 | −5.184 | Very well graded |
| Coefficient of curvature ($C_c$) | 0.364 | 1.459 | Poorly graded |
| Hydraulic Conductivity Coefficient ($K$) [cm/h] | 34.848 | | SP-SW (U.S.C.S) |
| | (cm/h) | | |
| Average Chart Size ($Mz$) | 1.866 | −0.900 | Small pebble, flow and average current energy |
| Inclusive graph standard deviation ($\Phi i$) | 0.202 | 2.308 | Very poorly selected. Low fluidity and high energy current |
| Degree of inclusive graphic asymmetry ($Ski$) | 11.349 | −0.183 | Asymmetrical towards thick |
| Measurement of graphical Kurtosis ($KG$) | 0.748 | 0.419 | Very platykurtic |
| Mode | 25.4, −4.75, −2, −0.08 | - | Multimodal: small pebble, granules and fine sand |
| Unified Soil Classification System (U.S.C.S.) | SP | | Well-graduated sands, sands with gravel, with few or no fines |
| | - | | - |
| **Samples code** | | LCHCP-18 | |
| **Statistical parameters** | **Opening (mm)** | **Aperture** | **Result** |
| | | **Phi ($\Phi$)** | |
| Coefficient of uniformity ($C_u$) | 45.000 | −5.492 | Very well graded |
| Coefficient of curvature ($C_c$) | 1.800 | −0.848 | Well gradado |
| Hydraulic Conductivity Coefficient ($K$) [cm/h] | 115.200 | | SP-SW (U.S.C.S) |
| | (cm/h) | | |
| Average Chart Size ($Mz$) | 5.528 | −2.467 | Small pebble, flow and average current energy |
| Inclusive graph standard deviation ($\Phi i$) | 0.239 | 2.065 | Very poorly selected. Low fluidity and high energy current |
| Degree of inclusive graphic asymmetry ($Ski$) | 0.91581265 | 0.127 | Asymmetrical towards thick |
| Measurement of graphical Kurtosis ($KG$) | 0.670 | 0.578 | Very platykurtic |
| Mode | 25.4, −9.5, −4.75, −0.08 | - | Multimodal: small pebble, granules and fine sand |
| Unified Soil Classification System (U.S.C.S.) | GW | Well-graded gravel, mixture of gravel and sand with few or no fines | |
| | GM | Silty gravels, poorly graduated mixtures of gravel, sand and silt | |

## Appendix I. Shape Parameters, Whose Value Is Related to Its Meaning

| Form Parameters | | | | | | | | |
|---|---|---|---|---|---|---|---|---|
| **Zone** | **A** | **P** | **L_A** | **A_P** | **AA** | **L_CP (km)** | **F** | **Kc** |
| | **(km²)** | **(km)** | **(km)** | **(km)** | **(km²)** | | | |
| La Chimba basin | 24.710 | 23.594 | 7.056 | 3.502 | 11.120 | 7.715 | 0.415 | 1.329 |
| La Chimba sub-basin | 17.217 | 21.000 | 7.056 | 2.440 | 7.748 | 7.715 | 0.289 | 1.417 |
| Guanaco sub-basin | 7.493 | 12.715 | 4.252 | 1.762 | 3.372 | 4.780 | 0.328 | 1.301 |

Notes: A. Area (km²). P. Perimeter (km). THE. Axial length (km). AP. Average width (km). PCL. Length of the main channel (km). F. Horton form factor (dimensionless). Kc. Compactness factor (dimensionless).

## Appendix J. Drainage Network Parameters, Whose Value Is Related to Their Meaning

| Drainage Network Parameters | | | | | |
|---|---|---|---|---|---|
| **Zona** | **Branch Relationship ($R_b$)** | **Length Ratio (RL)** | **Densidad de Drenajes ($D_d$)** | **Frecuencia de Drenajes ($F_d$)** | **Jerarquía de Drenajes (J)** |
| Cuenca La Chimba | 5.179 | 0.519 | 10.701 | 71.267 | 6 |
| | Mountain basins. Typical river system. Steep slopes with rapid runoff formation and minor flood maxims | May present high concentrations of runoff along the main channel | Very well drained basin. High volume generation runoff velocities | High probability that the rainwater drop will find a drain | |
| Subcuenca La Chimba | 5.729 | 0.547 | 10.925 | 70.744 | 5 |
| | Mountain basins. Typical river system. Steep slopes with rapid runoff formation and minor flood maxims | May present high concentrations of runoff along the main channel | Sub-basin very well drained. Large volume generation and runoff velocities | High probability that the rainwater drop will find a drain | |
| Subcuenca Guanaco | 4.747 | 0.572 | 10.188 | 72.467 | 5 |
| | Mountain basins. Typical river system. Steep slopes with rapid runoff formation and minor flood maxims | May present high concentrations of runoff along the main channel | Sub-basin very well drained. Large volume generation and runoff velocities | Very high probability that the rainwater droplet will find a drain | |

Notes: $R_b$. Bifurcation relation. RL. length ratio. $D_d$. Density of the drainage network. $F_d$. Drain frequency. J. Hierarchy of drains.

## Appendix K. Complementary Parameters, Whose Values Are Related to Their Meaning

| Supplementary Parameters | | |
|---|---|---|
| **Zone** | **Torrentiality Coefficient (Ct)** | **Potentiality Index (IP)** |
| La Chimba basin | 57.790 | 3.6 |
| La Chimba sub-basin | 56.862 | 5.092 |
| Guanaco sub-basin | 59.923 | 11.831 |

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
