# Peer review of "Flow-Type Landslides Analysis in Arid Zones: Application in La Chimba Basin in Antofagasta, Atacama Desert (Chile)"

_water, doi:10.3390/w14142225_

Round 1

Reviewer 1 Report

The manuscript titled “Flow-Type Landslides Analysis In Arid Areas: Application in the La Chimba basin in Antofagasta. Atacama Desert of Chile” aims to validate an articulate debris flow simulation by comparing the result with a historical event. As a whole, it sounds better as a Technical Note (or a Case Study) rather than an Article. However, it has several shortcomings that discourage its publication on Water in the present form.

 The manuscript starts with a wrong text and citation : “… from the year 2000 to 2019, 7,348 of them have been registered, increasing 39 by 174% compared to the 20 previous years (1980-1999)” is relative to all disasters (including drought, earthquake, volcanic activity, wildfire) as enumerated by the report “The Human Cost of Disasters - An overview of the last 20 years 2000-2019”. Please re-formulate the lines 38-44. The report is listed in References as [5]. However, also the style of the referenced report is not correct as well as others cited works (see below).

 Lines 50: what means “”torrentia5/27/2022l”? It seems that there are several textual errors. Please, check carefully throughout the text.

 Figure 2: what is the red triangle? Please report the explication in caption. Probably it is related to “the most flood-type landslide … occurred on June 191 …” (lines 73-74). But maybe not? However, I was not able to found the citation of this figure in the text. Please, remember to “insert your graphics (schemes, figures, etc.) in the main text after the paragraph of its first citation.” (see Instruction for Authors). For a further example, Figure 5 is not cited in the main text. And so on.

 English and language style are not appropriate throughout many paragraphs. It must be deeply improved. A first suggestion is to shorten and make sentences more understandable. There are problems especially with the paragraphs of lines 88-105, where it is necessary to rephrase the text. However, this is also true of almost the entire manuscript.

 Other sentences are very confusing. The data cannot be understood. For example: how large is the La Chimba basin? 101 or 24.7 square kilometres (see lines 108-112)?

 The manuscript lacks in pertinent scientific literature. As an example, given the centrality of the catastrophic event of 1991, because important works (such as Sepúlveda, S.; Rebolledo, S.; Easton, G. Recent catastrophic debris flows in Chile: Geological hazard, climatic relationships and human response. Quaternary International. 2006, 158. 83-95) are not quoted?

 One of the most important shortcomings is the inadequacy of the scientific literature on landslides in arid environments. As the authors know “extreme runoff triggered by heavy and intense precipitations has been widely described in deserts and drylands worldwide” (cit. from Moreiras, S.; Sepúlveda, S. (2021) Landslides in Arid and Semi-Arid Environments. In  Treatise on Geomorphology, 2nd Edition, Elsevier, Vol. 5 "Mass Movement Geomorphology", 322-337). Thus, the authors are invited to put their work in this broad research context and carefully consider it throughout discussion and conclusions.  

 A further shortage is having structured the manuscript as a case study without any contextualization at a broader level. The “threat of flow-type landslide” in arid regions is present in several worldwide areas. From this point of view, the manuscript can truly be qualified as a technical report of a Case Study. The tables with the description of the methods and equations are pertinent for a technical report, not for an article on a scientific journal. However, if they are deemed to be necessary, they should be moved to an Appendix.

 For the style of References, please carefully follow the Instruction for Authors. There are several shortcomings. For example, as before commented, the report “The Human Cost of Disasters - An overview of the last 20 years 2000-2019”, you listed as [5]. “UNDRR The human cost of disasters: an overview of the last 20 years (2000 – 2019); USA, 2019;” (line 644) is incomplete and imperfect. Who is the Publisher? How many pages does it consist of? And so on.

 Other quotes are also incomplete and inaccurate. For example the number [15], Los fenómenos de remoción en masa ocurridos en la región de Antofagasta en Junio 659 de 1991. Is it an article or a book? Again, for works not written in English it is advisable to specify the language (in Spanish, in this case). Please, carefully check the References list before the re-submission.

 Finally, I believe that the information about “… financial support from the Project 628 FONDEF ID 21I10181. … financial support to Centro de Investigación para 629 la Gestión Integrada del Riesgo de Desastres (CIGIDEN) Project Fondap ID 15110017” must be reported as “Funding”, not as “Acknowledgments”.

Author Response

Reviewer 1

Dear Reviewer, thank you very much for your comments and suggested corrections. The manuscript was considerably corrected, the introduction, order of the manuscript, etc. were changed a lot.
Below are the answers to your questions.

The manuscript starts with a wrong text and citation : “… from the year 2000 to 2019, 7,348 of them have been registered, increasing 39 by 174% compared to the 20 previous years (1980-1999)” is relative to all disasters (including drought, earthquake, volcanic activity, wildfire) as enumerated by the report “The Human Cost of Disasters - An overview of the last 20 years 2000-2019”. Please re-formulate the lines 38-44. The report is listed in References as [5]. However, also the style of the referenced report is not correct as well as others cited works (see below).

R.- Dear reviewer, you are right, this was fixed:

“On a global scale, climate change has caused enormous damage. Only from the year 2000 to 2019, 7,348 of them have been registered, increasing by 174% compared to the 20 previous years (1980-1999).” Lee et al., 2020 (DOI:10.5194/nhess-22-1577-2022), (UNDR, 2019). Of the various threats generated by natural phenomena, threats from intense precipitation are recurrent and on the rise. landslides are one of the most frequent geological hazards in mountainous regions. They are one of the natural hazards that cause the most severe damage to infrastructure and even cause deaths every year. They also represent the most relevant geomorpho-logical processes in the construction and modification of the relief.

“The Human Cost of Disasters - An overview of the last 20 years 2000-2019”. Please re-formulate the lines 38-44. The report is listed in References as [5].

R.- Dear Reviewer, this paragraph was reworded.

Lines 50: what means “”torrentia5/27/2022l”? It seems that there are several textual errors. Please, check carefully throughout the text.

R.- Corrected

 Figure 2: what is the red triangle? Please report the explication in caption. Probably it is related to “the most flood-type landslide … occurred on June 191 …” (lines 73-74). But maybe not? However, I was not able to found the citation of this figure in the text. Please, remember to “insert your graphics (schemes, figures, etc.) in the main text after the paragraph of its first citation.” (see Instruction for Authors). For a further example, Figure 5 is not cited in the main text. And so on.

R.- Dear Reviewer, the location of Figure 2 in the document was changed (below, after the text that mentions "the most flood-type landslide disaster in Chile... 1991" and also mentions the "red triangle" so that it is easy for readers to understand.

English and language style are not appropriate throughout many paragraphs. It must be deeply improved. A first suggestion is to shorten and make sentences more understandable. There are problems especially with the paragraphs of lines 88-105, where it is necessary to rephrase the text. However, this is also true of almost the entire manuscript.

R.- This was improved in the document

Other sentences are very confusing. The data cannot be understood. For example: how large is the La Chimba basin? 101 or 24.7 square kilometres (see lines 108-112)?

R.- This paragraph was corrected, and is now easier to understand:

“Specifically, the study area is in the northern sector of the city of Antofagasta, between the coordinates 23°30' and 23°34' South latitude and 70°17' and 70°24' West longitude; bounded to W by the coastal edge and to E by the trace of the Mititus fault, corresponding to the Atacama Fault System (Figure 3A and 3B). This area is made up of the La Chimba basin with an area of 24.7 km2 (Figure 3C), being considered one of the largest basins in the city of Antofagasta (Figure 1); in addition, according to Araya [34] and Chong et al., [35] there is previous evidence of activations, as in the hydrometeorological event of 1991. Despite the background presented, currently the alluvial deposits of the mouth of the La Chimba basin have a significant number of urbanizations, located adjacent to the La Chimba landfill and an aggregate extraction site. Likewise, real estate projects will be de-veloped, projecting a significant population increase. Considering that this basin is gain-ing importance, there are no studies that estimate the type of hydrological response to possible rainfall events, nor an estimate of the possible areas of impact for urban areas and their future growth, nor the consideration of flood mitigation works”

The manuscript lacks in pertinent scientific literature. As an example, given the centrality of the catastrophic event of 1991, because important works (such as Sepúlveda, S.; Rebolledo, S.; Easton, G. Recent catastrophic debris flows in Chile: Geological hazard, climatic relationships and human response. Quaternary International. 2006, 158. 83-95) are not quoted?

R.- Dear reviewer, the mentioned article was cited in the document. Indeed, it is a good reference for a good scientific article on this topic, and published in Web of Science. In addition, it was improved in the addition of other published manuscripts on the subject (papers).

A further shortage is having structured the manuscript as a case study without any contextualization at a broader level. The “threat of flow-type landslide” in arid regions is present in several worldwide areas. From this point of view, the manuscript can truly be qualified as a technical report of a Case Study. The tables with the description of the methods and equations are pertinent for a technical report, not for an article on a scientific journal. However, if they are deemed to be necessary, they should be moved to an Appendix.

R.- Dear reviewer, the Tables have been sent to Appendix, and improvements have been made as requested.

For the style of References, please carefully follow the Instruction for Authors. There are several shortcomings. For example, as before commented, the report “The Human Cost of Disasters - An overview of the last 20 years 2000-2019”, you listed as [5]. “UNDRR The human cost of disasters: an overview of the last 20 years (2000 – 2019); USA, 2019;” (line 644) is incomplete and imperfect. Who is the Publisher? How many pages does it consist of? And so on.

Other quotes are also incomplete and inaccurate. For example the number [15], Los fenómenos de remoción en masa ocurridos en la región de Antofagasta en Junio 659 de 1991. Is it an article or a book? Again, for works not written in English it is advisable to specify the language (in Spanish, in this case). Please, carefully check the References list before the re-submission.

R.- This was improved.

Finally, I believe that the information about “… financial support from the Project 628 FONDEF ID 21I10181. … financial support to Centro de Investigación para 629 la Gestión Integrada del Riesgo de Desastres (CIGIDEN) Project Fondap ID 15110017” must be reported as “Funding”, not as “Acknowledgments”.

R.- Dear Reviewer, by requirement of the Government of Chile, the government requires that the projects must be mentioned in the acknowledgments item, so that it can be valid at the time of publication. For example, in the manuscript from Chile that you previously mentioned in your reviews "Recent catastrophic debris flows in Chile: Geological hazard, climatic relationships and human response" funding was mentioned in acknowledgments. I hope you understand

Reviewer 2 Report

The manuscript is concerned with hydrometeorological analysis of flow-type landslide in a hyper-arid mountainous area, which is interesting. It is relevant and within the scope of the journal. However, the manuscript, in its present form, contains several weaknesses. Adequate revisions to the following points should be undertaken in order to justify recommendation for publication.

1.      The readability and presentation of the study should be further improved. The paper suffers from language problems.

2.      The main objective of this study is not clear. It’s more like a practical application of HEC-RAS and RAMMS. The authors should clearly emphasize the contribution of the study.

3.      The literature review seems very cursory. The importance of the design carried out in this manuscript can be explained better than other important studies published in this field. I recommend the authors to review other recently developed works.

4.      Section 1.2 study area should be merged into other sections in the form of a subsection.

5.      The quality of some figures (figs 1, 3, and 9) is low!

6.      Moreover, the manuscript could be substantially improved by relying and citing more on recent literatures in landslide monitoring and early warning such as the followings. http://dx.doi.org/10.1007/s00477-022-02183-5, https://doi.org/10.1155/2020/2624547. Discussions about result comparison and/or incorporation of those concepts in your works are encouraged:

7.      Line 50: torrentia5/27/2022l events. Please go through the manuscript for typos.

8.      Line 160: Intensity-Duration-Frequency has already been defined as IDF. Please go through the manuscript for abbreviations.

9.      English and Spanish are mixed in the manuscript. For example: Table 20: Parámetros del relieve. Fig 9. Please go through the manuscript to replace Spanish with English.

10.   There are many confusing statements which should be revised to enhance the text clarity.

11.   Lines 34-35: keywords overkill. There are nine keywords. Please merge.

Author Response

Dear Reviewer, thank you very much for your comments and suggested corrections. The manuscript was considerably corrected, the introduction, order of the manuscript, etc. were changed a lot.
Below are the answers to your questions.

  1. The readability and presentation of the study should be further improved. The paper suffers from language problems.

R.- This was improved

  1. The main objective of this study is not clear. It’s more like a practical application of HEC-RAS and RAMMS. The authors should clearly emphasize the contribution of the study.

R.-  Dear reviewer, the entire introductory framework of this manuscript was modified to make the study easier to understand. (I do not copy and paste the texts here because it would be too long. But you can review the new manuscript)

  1. The literature review seems very cursory. The importance of the design carried out in this manuscript can be explained better than other important studies published in this field. I recommend the authors to review other recently developed works.

R.- Dear Reviewer, many studies on the subject were also incorporated to improve the manuscript. Previously, many papers were missing to improve it.

  1. Section 1.2 study area should be merged into other sections in the form of a subsection.

R.- Dear Reviewer, this section has been improved and reorganized. In addition, many Tables in the manuscript were sent to Appendix

  1. The quality of some figures (figs 1, 3, and 9) is low!

R.- This was improved

  1. Moreover, the manuscript could be substantially improved by relying and citing more on recent literatures in landslide monitoring and early warning such as the followings. http://dx.doi.org/10.1007/s00477-022-02183-5, https://doi.org/10.1155/2020/2624547.Discussions about result comparison and/or incorporation of those concepts in your works are encouraged:

R.- Many new works were incorporated into the manuscript, it was generally improved by the use of these. In addition, one of these cites was incorporated.

  1. Line 50: torrentia5/27/2022l events. Please go through the manuscript for typos.

R.- Corrected

  1. Line 160: Intensity-Duration-Frequency has already been defined as IDF. Please go through the manuscript for abbreviations.

R.- Improved in the manuscript

  1. English and Spanish are mixed in the manuscript. For example: Table 20: Parámetros del relieve. Fig 9. Please go through the manuscript to replace Spanish with English.

R.- Corrected

  1. There are many confusing statements which should be revised to enhance the text clarity.

R.- Improved

  1. Lines 34-35: keywords overkill. There are nine keywords. Please merge.

R.- This was improved. Only 4 keywords left

Regards

Round 2

Reviewer 1 Report

“On a global scale, climate change has caused enormous damage. Only from the year 2000 to 2019, 7,348 of them have been registered, increasing by 174% compared to the 20 previous years (1980-1999) [1]” (lines 45-47, revised version). I appreciate the change but the new sentences are ambiguous. The 7,348 “enormous damage” (disaster events” in the original source, see below) were not all due to climate change. Instead such a number comprises also earthquakes, volcanic events, and other disasters not caused by climatic events (please see pages 6-7 of CRED and UNDRR: The human cost of disasters: an overview of the last 20 years (2000–2019), https://www.undrr.org/publication/human-cost-disasters-overview-last-20-years-2000-2019, you referenced as [18] in the revised version). However the authors now cite “Liu, L.; Gao, J.; Wu, S. Warming of 0.5 °C may cause double the economic loss and increase the population affected by floods 628 in China. Nat. Hazards Earth Syst. Sci. 2022”. These last authors correctly stated “Between 2000 and 2019, 7348 disaster events were recorded worldwide … with a surge in the number of climate-related disasters, of which floods were the most frequent, accounting for 44% of all disasters (CRED and UNDRR, 2020; WMO, 2021)”. We can conclude that most of the 7,348 disaster were climate-related. Please edit the incipit correctly.

Please considers that "Instructions for Authors" states "abstract should be a total of about 200 words maximum" (Abstract is now 395 words, 300 in the early version). Therefore, as a rule, the Abstract should be shortened. However, I defer to the editor decision.

About the “red triangle” in Figure 2, now is mentioned in the text (line 148). However, its meaning must also be explained in the caption of the figure.

As for the other comments, the authors modified the text more or less adequately. However, I still judge the manuscript as a Technical Note (or a Case Study) rather than an Article.

Author Response

Dear Reviewer

Hello, thank you very much for your corrections.

The correction requested in the introduction was made.

The abstract was reduced to 300 words

It was indicated in Figure 2 what is the red triangle

Regards

Reviewer 2 Report

The comments have been carefully addressed. Accept as it is.

Author Response

Dear Reviewer

Hello, thank you very much for your work and good comments on our manuscript.

Regards